# Generalisation in Lifelong Reinforcement Learning through Logical Composition

**Geraud Nangue Tasse, Steven James & Benjamin Rosman**
School of Computer Science and Applied Mathematics
University of the Witwatersrand
Johannesburg, South Africa
`geraudnt@gmail.com, {steven.james, benjamin.rosman1}@wits.ac.za`

## Abstract

We leverage logical composition in reinforcement learning to create a framework that enables an agent to autonomously determine whether a new task can be immediately solved using its existing abilities, or whether a task-specific skill should be learned. In the latter case, the proposed algorithm also enables the agent to learn the new task faster by generating an estimate of the optimal policy. Importantly, we provide two main theoretical results: we bound the performance of the transferred policy on a new task, and we give bounds on the necessary and sufficient number of tasks that need to be learned throughout an agent's lifetime to generalise over a distribution. We verify our approach in a series of experiments, where we perform transfer learning both after learning a set of base tasks, and after learning an arbitrary set of tasks. We also demonstrate that, as a side effect of our transfer learning approach, an agent can produce an interpretable Boolean expression of its understanding of the current task. Finally, we demonstrate our approach in the full lifelong setting where an agent receives tasks from an unknown distribution. Starting from scratch, an agent is able to quickly generalise over the task distribution after learning only a few tasks, which are sub-logarithmic in the size of the task space.

## 1 Introduction

Reinforcement learning (RL) is a framework that enables agents to learn desired behaviours by maximising the rewards received through interaction with an environment (Sutton et al., 1998). While RL has achieved recent success in several difficult, high-dimensional domains (Mnih et al., 2015; Levine et al., 2016; Lillicrap et al., 2016; Silver et al., 2017), these methods require millions of samples from the environment to learn optimal behaviours. This is ultimately a fatal flaw, since learning to solve complex, real-world tasks from scratch for *every* task of interest is typically infeasible. Hence a major challenge in RL is building general-purpose agents that are able to use existing knowledge to quickly solve new tasks in the environment. The question of interest is then: after learning $n$ tasks sampled from some distribution, how can an agent transfer or leverage the skills learned from those $n$ tasks to improve its starting performance or learning speed in task $n + 1$?

This problem setting is formalised by lifelong RL (Thrun, 1996; Abel et al., 2018). One approach to transfer in lifelong RL is *composition* (Todorov, 2009), which allows an agent to leverage its existing skills to build complex, novel behaviours that can then be used to solve or speed up learning of a new task (Todorov, 2009; Saxe et al., 2017; Haarnoja et al., 2018; van Niekerk et al., 2019; Hunt et al., 2019; Peng et al., 2019). Recently, Nangue Tasse et al. (2020) proposed a framework for defining a Boolean algebra over the space of tasks and their optimal value functions. This allowed for tasks and value functions to be composed using the union, intersection and negation operators in a principled manner to yield optimal skills zero-shot.

In this work, we propose a framework for lifelong RL that focuses not only on transfer between tasks for faster RL, but also provides guarantees on the generalisation of an agent's skills over an unknown task distribution. We first extend the logical composition framework of Nangue Tasse et al. (2020) to discounted and stochastic tasks. We provide theoretical bounds for our approach in stochastic settings,

and also compare them to previous work in the discounted setting. We then show how our framework leverages logical composition to tackle the lifelong RL problem. The framework enables agents to iteratively solve tasks as they are given, while at the same time constructing a *library* of skills that can be composed to obtain behaviours for solving future tasks faster, or even without further learning.

We empirically verify our framework in a series of experiments, where an agent is i) pretrained on a set of base tasks provided by the Boolean algebra framework, and ii) when the pretrained tasks are not base tasks. We show that agents here are able to achieve significant jumpstarts on new tasks. Finally, we demonstrate our framework in the lifelong RL setting where an agent receives tasks from an unknown (possibly non-stationary) distribution and must determine what skills to learn and add to its library, and how to combine its current skills to solve new tasks. Results demonstrate that this framework enables agents to quickly learn a set of skills, resulting in a combinatorial explosion in their abilities. Consequently, even when tasks are sampled randomly from an unknown distribution, an agent can leverage its existing skills to solve new tasks without further learning, thereby generalising over task distributions.

## 2 BACKGROUND

We consider tasks modelled by Markov Decision Processes (MDPs). An MDP is defined by the tuple $(\mathcal{S}, \mathcal{A}, p, r, \gamma)$, where (i) $\mathcal{S}$ is the state space, (ii) $\mathcal{A}$ is the action space, (iii) $p(s'|s, a)$ is a Markov transition probability, (iv) $r$ is the real-valued reward function bounded by $[r_{\text{MIN}}, r_{\text{MAX}}]$, and (v) $\gamma \in [0, 1)$ is the discount factor. In this work, we focus on tasks where an agent is required to reach a set of desirable goals in a goal space $\mathcal{G} \subseteq \mathcal{S}$ (a set of boundary states). Here, termination in $\mathcal{G}$ is modelled similarly to van Niekerk et al. (2019) by augmenting the state space with a virtual state, $\omega$, such that $p(\omega|s, a) = 1 \ \forall (s, a) \in (\mathcal{G} \times \mathcal{A})$ and the rewards are zero after reaching $\omega$. We hence consider the set of tasks $\mathcal{M}$ such that the tasks are in the same environment—described by a background MDP $(\mathcal{S}, \mathcal{A}, p, \gamma, r_0)$—and each task can be uniquely specified by a set of desirable and undesirable goals:

$$\mathcal{M}(\mathcal{S}, \mathcal{A}, p, \gamma, r_0) := \{(\mathcal{S}, \mathcal{A}, p, \gamma, r) \mid \forall a \in \mathcal{A}, \ r(s, a) = r_0(s, a) \ \forall s \in \mathcal{S} \setminus \mathcal{G};$$
$$r(g, a) = r_g \in \{r_{\text{MIN}}, r_{\text{MAX}}\} \ \forall g \in \mathcal{G}\} \quad (1)$$

The goal of the agent is to compute a Markov policy $\pi$ from $\mathcal{S}$ to $\mathcal{A}$ that optimally solves a given task. A given policy $\pi$ is characterised by a value function $V^\pi(s) = \mathbb{E}_\pi \left[\sum_{t=0}^{\infty} \gamma^t r(s_t, a_t)\right]$, specifying the expected return obtained under $\pi$ starting from state $s$. The *optimal* policy $\pi^*$ is the policy that obtains the greatest expected return at each state: $V^{\pi^*}(s) = V^*(s) = \max_\pi V^\pi(s)$ for all $s$ in $\mathcal{S}$. A related quantity is the $Q$-value function, $Q^\pi(s, a)$, which defines the expected return obtained by executing $a$ from $s$, and thereafter following $\pi$. Similarly, the optimal $Q$-value function is given by $Q^*(s, a) = \max_\pi Q^\pi(s, a)$ for all $s$ in $\mathcal{S}$ and $a$ in $\mathcal{A}$.

### 2.1 LOGICAL COMPOSITION

Nangue Tasse et al. (2020) recently proposed the notion of a Boolean task algebra, which allows an agent to perform logical operations—conjunction ($\wedge$), disjunction ($\vee$) and negation ($\neg$)—over the space of tasks and value functions. While they only considered deterministic shortest path tasks ($\gamma = 1$ with deterministic dynamics), we summarise their approach here and later extend it to discounted stochastic tasks (Section 3.1).

To achieve zero-shot logical composition, Nangue Tasse et al. (2020) extend the standard rewards and value functions used by an agent to define goal-oriented versions as follows:

**Definition 1.** *The extended reward function $\bar{r} : \mathcal{S} \times \mathcal{G} \times \mathcal{A} \to \mathbb{R}$ is given by the mapping*

$$(s, g, a) \mapsto \begin{cases} \bar{r}_{MIN} & \text{if } g \neq s \text{ and } s \in \mathcal{G} \\ r(s, a) & \text{otherwise,} \end{cases} \quad (2)$$

*where $\bar{r}_{MIN} \leq \min\{r_{MIN}, (r_{MIN} - r_{MAX})D\}$, and $D$ is the diameter of the MDP (Jaksch et al., 2010).*

**Definition 2.** *The extended $Q$-value function $\bar{Q} : \mathcal{S} \times \mathcal{G} \times \mathcal{A} \to \mathbb{R}$ is given by the mapping*

$$(s, g, a) \mapsto \bar{r}(s, g, a) + \gamma \sum_{s' \in \mathcal{S}} p(s'|s, a) \bar{V}^{\bar{\pi}}(s', g), \quad (3)$$

where $\bar{V}^{\bar{\pi}}(s, g) = \mathbb{E}_{\bar{\pi}}\left[\sum_{t=0}^{\infty} \bar{r}(s_t, g, a_t)\right]$.

By penalising the agent for achieving goals different from those it wanted to reach ($\bar{r}_{MIN}$ if $g \neq s$ and $s \in \mathcal{G}$), the extended reward function has the effect of driving the agent to learn how to separately achieve all desirable goals. Importantly, the standard reward and value functions can be recovered from their extended versions by simply maximising over goals. As such, the agent can also recover the task policy by maximising over both goals and actions: $\pi(s) \in \arg\max_{a \in \mathcal{A}} \max_{g \in \mathcal{G}} \bar{Q}(s, g, a)$.

The logic operators over tasks and extended action-value functions are then defined as follows:

**Definition 3.** *Let $\mathcal{M}$ be a set of tasks with bounds $\mathcal{M}_{MIN}, \mathcal{M}_{MAX} \in \mathcal{M}$ such that,*

$$r_{\mathcal{M}_{MAX}}(s, a) := \max_{M \in \mathcal{M}} r_M(s, a) \qquad r_{\mathcal{M}_{MIN}}(s, a) := \min_{M \in \mathcal{M}} r_M(s, a)$$

*Define the $\neg, \vee$, and $\wedge$ operators over $\mathcal{M}$ as*

$$\neg(M) := (\mathcal{S}, \mathcal{A}, p, r_{\neg M}), \text{ where } r_{\neg M}(s, a) := (r_{\mathcal{M}_{MAX}}(s, a) + r_{\mathcal{M}_{MIN}}(s, a)) - r_M(s, a)$$

$$\vee(M_1, M_2) := (\mathcal{S}, \mathcal{A}, p, r_{M_1 \vee M_2}), \text{ where } r_{M_1 \vee M_2}(s, a) := \max\{r_{M_1}(s, a), r_{M_2}(s, a)\}$$

$$\wedge(M_1, M_2) := (\mathcal{S}, \mathcal{A}, p, r_{M_1 \wedge M_2}), \text{ where } r_{M_1 \wedge M_2}(s, a) := \min\{r_{M_1}(s, a), r_{M_2}(s, a)\}$$

**Definition 4.** *Let $\bar{\mathcal{Q}}^*$ be the set of optimal extended $\bar{Q}$-value functions for tasks in $\mathcal{M}$, with bounds $\bar{Q}_{MIN}^*, \bar{Q}_{MAX}^* \in \bar{\mathcal{Q}}^*$ which are respectively the optimal $\bar{Q}$-functions for the tasks $\mathcal{M}_{MIN}, \mathcal{M}_{MAX} \in \mathcal{M}$. Define the $\neg, \vee$, and $\wedge$ operators over $\bar{\mathcal{Q}}^*$ as,*

$$\neg(\bar{Q}^*)(s, g, a) := \left(\bar{Q}_{MIN}^*(s, g, a) + \bar{Q}_{MAX}^*(s, g, a)\right) - \bar{Q}^*(s, g, a)$$

$$\vee(\bar{Q}_1^*, \bar{Q}_2^*)(s, g, a) := \max\{\bar{Q}_1^*(s, g, a), \bar{Q}_2^*(s, g, a)\}$$

$$\wedge(\bar{Q}_1^*, \bar{Q}_2^*)(s, g, a) := \min\{\bar{Q}_1^*(s, g, a), \bar{Q}_2^*(s, g, a)\}$$

Using the definitions for the logical operations over $\mathcal{M}$ and $\bar{\mathcal{Q}}^*$ given above, Nangue Tasse et al. (2020) construct a Boolean algebra over tasks and extended value functions. Furthermore, by leveraging the goal-oriented definition of extended value functions, they also show that $\mathcal{M}$ and $\bar{\mathcal{Q}}^*$ are homomorphic. As a result, if a task can be expressed using the Boolean algebra, the optimal value function for the task can immediately be computed. This enables agents to solve any new task that is given as the logical combination of learned ones.

## 3  LIFELONG TRANSFER THROUGH COMPOSITION

In lifelong RL, an agent is presented with a series of tasks sampled from some distribution $\mathcal{D}$. The agent then needs to not only transfer knowledge learned from previous tasks to solve new but related tasks quickly, but it also should not forget learned knowledge in the process. We formalise this lifelong learning problem as follows:

**Definition 5.** *Let $\mathcal{D}$ be an unknown, possibly non-stationary, distribution over a set of tasks $\mathcal{M}(\mathcal{S}, \mathcal{A}, p, \gamma, r_0)$. The lifelong learning problem consists of the repetition of the following steps for $t \in \mathbb{N}$:*

> *1. The agent is given a task $M_t \sim \mathcal{D}(t)$,*
>
> *2. The agent interacts with the MDP $M_t$ until it is $\epsilon$-optimal in $M_0, ..., M_t$.*

This formulation of lifelong RL is similar to that of Abel et al. (2018); the main difference is that we do not assume that $\mathcal{D}$ is stationary, and we explicitly require an agent to retain learned skills.

As discussed in the introduction, one of the main goals in this setting is that of transfer (Taylor & Stone, 2009). We add an important question to this setting: how many tasks should an agent learn

during its lifetime in order to generalise over the task distribution? In other words, how many tasks should it learn to be able to solve any new task immediately? While most approaches focus on the goal of transfer, the question of the number of tasks is often neglected by simply assuming the case where the agent has already learned $n$ tasks (Abel et al., 2018; Barreto et al., 2018). Consider, for example, a task space with only $|\mathcal{G}| = 40$ goals. Then, given the combination of all possible goals, the size of the task space is $|\mathcal{M}| = 2^{|\mathcal{G}|} \approx 10^{12}$. If $\mathcal{D}$ is a uniform distribution over $|\mathcal{M}|$, then for most transfer learning methods an agent will have to learn most of the tasks it is presented with, since the probability of observing the same task will be approximately zero. This is clearly impractical for a setting like RL, where learning methods often have a high sample complexity even with transfer learning. It is also extremely memory inefficient, since the learned skills of most tasks must be stored.

## 3.1 EXTENDING THE BOOLEAN ALGEBRA FRAMEWORK

In this section, we show how logical composition can be leveraged to learn a subset of tasks that is sufficient to generalise over the task distribution. Since the logical composition results of Nangue Tasse et al. (2020) were only shown for deterministic shortest path tasks (where $\gamma = 1$), we extend the framework to discounted and stochastic tasks $\mathcal{M}$ (Equation 1). To achieve this, we first redefine the extended reward function (Definition 1) to use the simpler penalty $\bar{r}_{MIN} = r_{\text{MIN}}$. We also redefine $\neg$ over $\bar{\mathcal{Q}}^*$ as follows:

$$\neg(\bar{Q}^*)(.) := \begin{cases} \bar{Q}^*_{MAX}(.) & \text{if } |\bar{Q}^*(.) - \bar{Q}^*_{MIN}(.)| \leq |\bar{Q}^*(.) - \bar{Q}^*_{MAX}(.)| \\ \bar{Q}^*_{MIN}(.) & \text{otherwise,} \end{cases}, \; \forall(.) \in \mathcal{S} \times \mathcal{G} \times \mathcal{A}.$$

The intuition behind this re-definition of the negation operator is as follows: since each goal is either desirable or not, the optimal extended value function $\bar{Q}^*(s, g, a)$ is either $\bar{Q}^*_{MAX}(s, g, a)$ or $\bar{Q}^*_{MIN}(s, g, a)$. Hence, if $\bar{Q}^*(s, g, a)$ is closer to $\bar{Q}^*_{MIN}(s, g, a)$, then its negation should be $\bar{Q}^*_{MAX}(s, g, a)$, and vice versa. For tasks in $\mathcal{M}$, this is equivalent to the previous definition of $\neg$ for optimal $\bar{Q}$-value functions, but it will give us tight bounds when composing $\epsilon$-optimal $\bar{Q}$-value functions (see Theorem 1).

We now show that the Boolean algebra and zero-shot composition results of Nangue Tasse et al. (2020) also hold for tasks in $\mathcal{M}$.

**Proposition 1.** *Let $\bar{\mathcal{Q}}^*$ be the set of optimal $\bar{Q}$-value functions for tasks in $\mathcal{M}$. Let $\mathscr{A} : \mathcal{M} \to \bar{\mathcal{Q}}^*$ be any map from $\mathcal{M}$ to $\bar{\mathcal{Q}}^*$ such that $\mathscr{A}(M) = \bar{Q}^*_M$ for all $M$ in $\mathcal{M}$. Then,*

  *(i) $\mathcal{M}$ and $\bar{\mathcal{Q}}^*$ respectively form a Boolean task algebra $(\mathcal{M}, \vee, \wedge, \neg, \mathcal{M}_{MAX}, \mathcal{M}_{MIN})$ and a Boolean extended value functions algebra $(\bar{\mathcal{Q}}^*, \vee, \wedge, \neg, \bar{Q}^*_{MAX}, \bar{Q}^*_{MIN})$,*

  *(ii) $\mathscr{A}$ is a homomorphism between $\mathcal{M}$ and $\bar{\mathcal{Q}}^*$.*

We can now solve any new task in $\mathcal{M}$ zero-shot if we are given the correct Boolean expression that informs the agent how to compose its optimal skills. This is essential for the following results.

## 3.2 TRANSFER BETWEEN TASKS

In this section, we leverage the logical composition results to address the following question of interest: given an arbitrary set of learned tasks, can we transfer their skills to solve new tasks faster? As we will show in Theorem 1, we answer this question in the affirmative. To achieve this, we first note that each task $M \in \mathcal{M}$ can be associated with a binary vector $T \in \{0, 1\}^{|\mathcal{G}|}$ which represents its set of desirable goals, as illustrated by the tasks in Table 1. The approximation $\tilde{T}$ of this task representation can be learned just from task rewards ($r_M(s, a)$) by simply computing $\tilde{T}(s) = \mathbf{1}_{r_M(s,a)=r_{MAX}}$ at each terminal state $s$ that the agent reaches. We can then use any generic method, such as the *sum-of-products (SOP)*, to determine a candidate Boolean expression ($\mathcal{B}_{EXP}$) in terms of the learned binary representations $\tilde{\mathcal{T}}_n = \{\tilde{T}_1, ..., \tilde{T}_n\}$ of a set of past tasks $\hat{\mathcal{M}} = \{M_1, ..., M_n\} \subseteq \mathcal{M}$. An estimate of the optimal $\bar{Q}$-value function of $M$ can then be obtained by composing the learned $\bar{Q}$-value functions $\tilde{\bar{\mathcal{Q}}}^*_n = \{\tilde{\bar{Q}}^*_1, ..., \tilde{\bar{Q}}^*_n\}$ according to $\mathcal{B}_{EXP}$. Theorem 1 shows the optimality of this process.[1]

---

[1] See Appendix A for proofs of theorems and Appendix B for a brief description of the $SOP$ method.

**Theorem 1.** *Let $M \in \mathcal{M}$ be a task with reward function $r$, binary representation $T$ and optimal extended action-value function $\bar{Q}^*$. Given $\epsilon$-approximations of the binary representations $\tilde{\mathcal{T}}_n = \{\tilde{T}_1, ..., \tilde{T}_n\}$ and optimal $\bar{Q}$-functions $\tilde{\bar{\mathcal{Q}}}^*_n = \{\tilde{\bar{Q}}^*_1, ..., \tilde{\bar{Q}}^*_n\}$ for $n$ tasks $\hat{\mathcal{M}} = \{M_1, ..., M_n\} \subseteq \mathcal{M}$, let*

$$T_{\mathcal{F}} = \mathcal{B}_{EXP}(\tilde{\mathcal{T}}_n) \text{ and } \bar{Q}_{\mathcal{F}} = \mathcal{B}_{EXP}(\tilde{\bar{\mathcal{Q}}}^*_n),$$

*where $\mathcal{B}_{EXP}$ is derived from $\tilde{\mathcal{T}}_n$ and $\tilde{T}$ using a generic method $\mathcal{F}$. Define $\pi(s) \in \arg\max_{a \in \mathcal{A}} Q_{\mathcal{F}}$ where $Q_{\mathcal{F}} := \max_{g \in \mathcal{G}} \bar{Q}_{\mathcal{F}}(s, g, a)$. Then,*

*(i)* $\|Q^* - Q^\pi\|_\infty \leq \frac{2}{1-\gamma}((\mathbf{1}_{T \neq T_{\mathcal{F}}} + \mathbf{1}_{r \notin \{r_g\}_{|\mathcal{G}|}})r_\Delta + \epsilon),$

*(ii) if the dynamics are deterministic,*

$$\|Q^* - Q_{\mathcal{F}}\|_\infty \leq (\mathbf{1}_{T \neq T_{\mathcal{F}}})r_\Delta + \epsilon,$$

*where $\mathbf{1}$ is the indicator function, $r_g(s, a) := \bar{r}(s, g, a)$, $r_\Delta := r_{MAX} - r_{MIN}$, and $\|f - h\|_\infty := \max_{s,g,a} |f(s, g, a) - h(s, g, a)|$.*

Theorem 1(i) states that if $\bar{Q}_{\mathcal{F}}$ is close to optimal, then acting greedily with respect to it is also close to optimal. Interestingly, this is similar to the bound obtained by Barreto et al. (2018) (Proposition 1) for transfer learning using generalised policy improvement (GPI), but stronger.[2] This is unsurprising, since $\pi(s) \in \arg\max_{a \in \mathcal{A}} \max_{g \in \mathcal{G}} \bar{Q}_{\mathcal{F}}(s, g, a)$ can be interpreted as generalised policy improvement on the set of goal policies of the extended value function $\bar{Q}_{\mathcal{F}}$. Importantly, if the environment is deterministic, then we obtain a strong bound on the composed value functions (Theorem 1(ii)). This bound shows that transfer learning using logical composition is $\epsilon$-optimal—that is, there is no loss in optimality—when the new task is expressible as a logical combination of past ones. With the exponential nature of logical combination, this gives agents a strong generalisation ability over the task space—and hence over any task distribution—as we will show in Theorem 2.

### 3.3 GENERALISATION OVER A TASK DISTRIBUTION

We leverage Theorem 1 to design an algorithm that combines the $SOP$ approach with goal-oriented learning to achieve fast transfer in lifelong RL. Given an off-policy RL algorithm $\mathscr{A}$, the agent initializes its extended value function $\tilde{\bar{Q}}$, the task binary vector $\tilde{T}$, and a goal buffer. At the beginning of each episode, the agent computes $T_{SOP}$ and $Q_{SOP}$ for $\tilde{T}$ using the $SOP$ method and its library of learned task vectors and extended Q-functions. It then acts using the behaviour policy ($\epsilon$-greedy for example) of $\mathscr{A}$ with $\bar{Q}_{SOP}$ for the action-value function if $T_{SOP} = \tilde{T}$, and $\bar{Q}_{SOP} \vee \tilde{\bar{Q}}$ otherwise.[3] If $T_{SOP} \neq \tilde{T}$, the agent also updates $\tilde{\bar{Q}}$ for each goal in the goal buffer using $\mathscr{A}$. Additionally, when the agent reaches a terminal state $s$, it adds it to the goal buffer and updates $\tilde{T}(s)$ using the reward it receives ($\tilde{T}(s) = \mathbf{1}_{r_M(s,a)=r_{MAX}}$). Training stops when the agent has reached the desired level of optimality (or after $n$ episodes in practice), after which the agent adds the learned $\tilde{T}$ and $\tilde{\bar{Q}}$ to its library if $T_{SOP} \neq \tilde{T}$. The full algorithm is included in Appendix B. We refer to this algorithm as SOPGOL (*Sum Of Products with Goal-Oriented Learning*).

When $\mathcal{G}$ is finite, we show in Theorem 2 that SOPGOL generalises over any unknown task distribution after learning only a number of tasks logarithmic in the size of the task space. The lower bound is $\lceil log|\mathcal{G}| \rceil$, since this is the minimum number of tasks that span the task space, as can be seen in Table 1 (top) for example. The upper bound is $|\mathcal{G}|$ because that is the dimensionality of the task binary representations $\{0, 1\}^{|\mathcal{G}|}$. Since the number of tasks is $|\mathcal{M}| = 2^{|\mathcal{G}|}$, we have that the upper bound $|\mathcal{G}| = log|\mathcal{M}|$ is logarithmic in the size of the task space.

---

[2]See Section 1.4 of the appendix for a detailed discussion of this with the simplification of the bound in Proposition 1 (Barreto et al., 2018) to the same form as Theorem 1(i).

[3]Since $\bar{Q}_{SOP} \vee \tilde{\bar{Q}} = \max\{\bar{Q}_{SOP}, \tilde{\bar{Q}}\}$, it is equivalent to GPI and hence is guaranteed to be equal or more optimal than the individual value functions. Hence using $\bar{Q}_{SOP} \vee \tilde{\bar{Q}}$ in the behaviour policy gives a straightforward way of leveraging $\bar{Q}_{SOP}$ to learn $\tilde{\bar{Q}}$ faster.

**Theorem 2.** *Let $\mathcal{D}$ be an unknown, possibly non-stationary, distribution over a set of tasks $\mathcal{M}(\mathcal{S}, \mathcal{A}, p, \gamma, r_0)$ with finite $\mathcal{G}$. Let $\mathscr{A} : \mathcal{M} \to \bar{\mathcal{Q}}^*$ be any map from $\mathcal{M}$ to $\bar{\mathcal{Q}}^*$ such that $\mathscr{A}(M) = \bar{Q}_M^*$ for all $M$ in $\mathcal{M}$. Let*

$$\tilde{\mathcal{T}}_{t+1}, \tilde{\bar{\mathcal{Q}}}_{t+1}^* = SOPGOL(\mathscr{A}, M_t, \tilde{\mathcal{T}}_t, \tilde{\bar{\mathcal{Q}}}_t^*) \text{ where } M_t \sim \mathcal{D}(t) \text{ and } \tilde{\mathcal{T}}_0 = \tilde{\bar{\mathcal{Q}}}_0^* = \varnothing \; \forall t \in \mathbb{N}.$$

*Then,*

$$\lceil \log |\mathcal{G}| \rceil \le \lim_{t \to \infty} N_t \le |\mathcal{G}| \quad \text{where } N_t := |\tilde{\mathcal{T}}_t| = |\tilde{\bar{\mathcal{Q}}}_t^*|.$$

Interestingly, Theorem 2 holds even in the case where a new task is expressible in terms of past tasks ($T_{SOP} = \tilde{T}$), but we wish to solve it to a higher degree of optimality than past tasks. In this case, we can pretend $T_{SOP} \ne \tilde{T}$ and learn a new $\bar{Q}$-function to the desired degree of optimality. We can then add it to our library, and remove any other skill from our library (the least optimal for example).

## 4 EXPERIMENTS

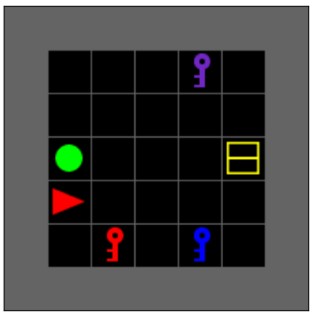

Figure 1: PICKUPOBJ domain. The red triangle represents the agent.

Table 1: Binary representation for base (top) and test (bottom) tasks. **0** or **1** corresponds to a goal reward of $r_{\text{MIN}}$ or $r_{\text{MAX}}$.

| Goals | 🔑 | ● | ▤ | 🔑 | ● | ▤ | 🔑 | ● | ▤ | 🔑 | ● | ▤ | 🔑 | ○ | ▤ |
|---|---|---|---|---|---|---|---|---|---|---|---|---|---|---|---|
| $T_a$ | 1 | 0 | 1 | 0 | 1 | 0 | 1 | 0 | 1 | 0 | 1 | 0 | 1 | 0 | 1 |
| $T_b$ | 0 | 1 | 1 | 0 | 0 | 1 | 1 | 0 | 0 | 1 | 1 | 0 | 0 | 1 | 1 |
| $T_c$ | 0 | 0 | 0 | 1 | 1 | 1 | 1 | 0 | 0 | 0 | 0 | 1 | 1 | 1 | 1 |
| $T_d$ | 0 | 0 | 0 | 0 | 0 | 0 | 0 | 1 | 1 | 1 | 1 | 1 | 1 | 1 | 1 |

| Goals | 🔑 | ● | ▤ | 🔑 | ● | ▤ | 🔑 | ● | ▤ | 🔑 | ● | ▤ | 🔑 | ○ | ▤ |
|---|---|---|---|---|---|---|---|---|---|---|---|---|---|---|---|
| $T_1$ | 0 | 0 | 0 | 0 | 0 | 0 | 0 | 0 | 0 | 0 | 0 | 0 | 0 | 0 | 1 |
| $T_2$ | 0 | 0 | 1 | 0 | 1 | 1 | 1 | 0 | 1 | 1 | 0 | 0 | 1 | 0 | 0 |
| $T_3$ | 1 | 1 | 0 | 1 | 0 | 0 | 0 | 1 | 0 | 1 | 1 | 0 | 0 | 1 | 1 |

### 4.1 TRANSFER AFTER PRETRAINING ON A SET OF TASKS

We consider the PICKUPOBJ domain from the MINIGRID environment (Chevalier-Boisvert et al., 2018), illustrated by Figure 1, where an agent must navigate in a 2D room to pick up objects of various shapes and colours from pixel observations.[4] This type of domain is prototypical in the literature (Nangue Tasse et al., 2020; Barreto et al., 2020; van Niekerk et al., 2019; Abel et al., 2018), because it allows for easy demonstration of transfer learning in many-goal tasks. In this domain, there are $|\mathcal{G}| = 15$ goals each corresponding to picking up objects of 3 possible types—box, ball, key—and 5 possible colours—red, blue, green, purple, and yellow. Hence a set of $\lceil \log_2 |\mathcal{G}| \rceil = 4$ base tasks can be selected that can be used to solve all $2^{|\mathcal{G}|} = 32768$ possible tasks under a Boolean composition of goals. The agent receives a reward of 2 when it picks up desired objects, and $-0.1$ otherwise. For all of our experiments in this section, we use deep Q-learning (Mnih et al., 2015) as the RL method for SOPGOL and as the performance baseline. We also compare SOPGOL to SOPGOL-transfer, or to SOPGOL-continual. SOPGOL-transfer refers to when no new skill is learned and SOPGOL-continual refers to when a new skill is always learned using the $SOP\ Q$ estimate to speed up learning, even if the new task could be solved zero shot. Since SOPGOL determines automatically which one to use, we compare whichever one it chooses with the other one in each of our experiments.

We first demonstrate transfer learning after pretraining on a set of base tasks—a minimal set of tasks that span the task space. This can be done if the set of goals is known upfront, by first assigning a Boolean label to each goal in a table and then using the rows of the table as base tasks. These are illustrated in Table 1 (top). Having learned the $\epsilon$-optimal extended value functions for our base tasks,

---

[4]Further environment details are given in Appendix D.

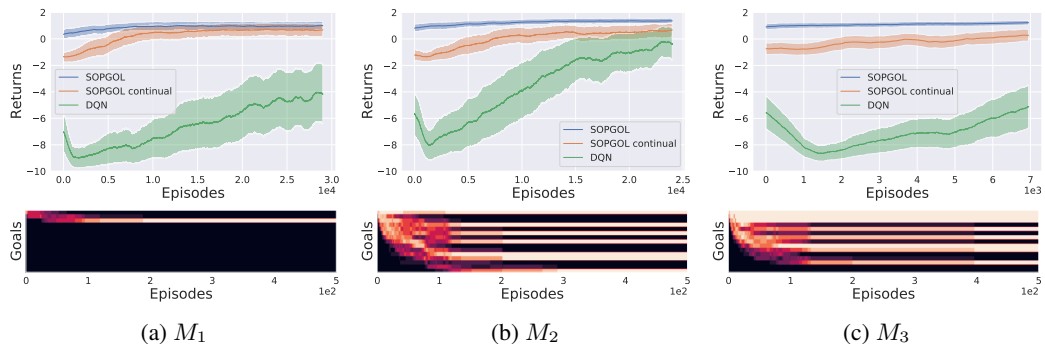

(a) $M_1$        (b) $M_2$        (c) $M_3$

Figure 2: Episodic returns (top) and learned binary representations (bottom) for test tasks $M_1$, $M_2$ and $M_3$ after pretraining on the base set of tasks $M_a$, $M_b$, $M_c$ and $M_d$. The shaded regions on the episodic returns indicate one standard deviation over 4 runs. The learned binary representations are similarly averaged over 4 runs, and reported for the first 500 episodes. The initial drop in DQN performance is as a result of the initial exploration phase where the exploration constant decays from 0.5 to 0.05. The Boolean expressions generated by SOPGOL during training for the respective test tasks are:

$$M_1 = M_a \wedge M_b \wedge M_c \wedge M_d,$$
$$M_2 = (M_a \wedge \neg M_b \wedge \neg M_d) \vee (M_a \wedge M_c \wedge M_d) \vee (\neg M_a \wedge M_b \wedge \neg M_c \wedge \neg M_d) \vee (\neg M_a \wedge \neg M_b \wedge \neg M_c \wedge M_d),$$
$$M_3 = (M_a \wedge M_b \wedge M_c) \vee (M_a \wedge \neg M_b \wedge \neg M_d) \vee (M_a \wedge M_c \wedge M_d) \vee (\neg M_a \wedge M_b \wedge \neg M_c \wedge \neg M_d) \vee (\neg M_a \wedge \neg M_b \wedge \neg M_c \wedge M_d) \vee (\neg M_b \wedge M_c \wedge \neg M_d).$$

we can now leverage logical composition for transfer learning on test tasks. We consider the three test tasks shown in Table 1 (bottom). For each, we run SOPGOL, SOPGOL-continual, and a standard DQN. Figure 2 illustrates the results where, as predicted by our theoretical results in Section 3.2, SOPGOL correctly determines that the current test tasks are solvable from the logical combinations of the learned base tasks. Its performance from the start of training is hence the best.

Now that we have demonstrated how SOPGOL enables an agent to solve any new task in an environment after training on base tasks, we consider the more practical case where new tasks are not fully expressible as a Boolean expression of previously learned tasks. The agent in this case

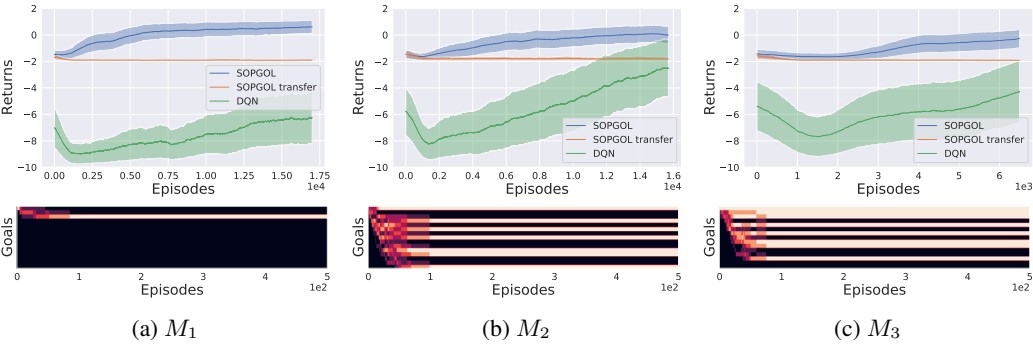

(a) $M_1$        (b) $M_2$        (c) $M_3$

Figure 3: Episodic returns (top) and learned binary representations (bottom) for test tasks $M_1$, $M_2$ and $M_3$ after pretraining on the non-base set of tasks ■,■,■ and ⌦. The shaded regions on the episodic returns indicate one standard deviation over 4 runs. The learned binary representations are similarly averaged over 4 runs, and reported for the first 500 episodes. The initial drop in DQN performance is a result of the initial exploration phase where the exploration constant decays from 0.5 to 0.05. The Boolean expressions generated by SOPGOL for the respective test tasks are:

$$\widetilde{M_1} = \neg \blacksquare \wedge \neg \blacksquare \wedge \blacksquare \wedge \neg \text{⌦},$$
$$\widetilde{M_2} = (\blacksquare \wedge \neg \blacksquare \wedge \neg \blacksquare) \vee (\neg \blacksquare \wedge \blacksquare \wedge \neg \blacksquare \wedge \text{⌦}) \vee (\neg \blacksquare \wedge \neg \blacksquare \wedge \blacksquare \wedge \text{⌦}) \vee (\neg \blacksquare \wedge \neg \blacksquare \wedge \neg \text{⌦}),$$
$$\widetilde{M_3} = (\neg \blacksquare \wedge \neg \blacksquare \wedge \neg \text{⌦}) \vee (\neg \blacksquare \wedge \neg \blacksquare) \vee (\neg \blacksquare \wedge \neg \blacksquare \wedge \neg \text{⌦}).$$

is pretrained on a set of tasks that do not span the task space, $\{\blacksquare, \blacksquare, \blacksquare, \mathbb{P}\}$, corresponding to the tasks of picking up green objects, blue objects, yellow objects, and keys. We then train the agent with SOPGOL, SOPGOL-transfer, and a standard DQN on the same set of test tasks considered previously (Table 1 (bottom)). The results in Figure 3 demonstrate how SOPGOL now chooses to learn a task-specific skill after transfer, and hence outperforms SOPGOL-transfer since the test tasks are not entirely expressible in terms of the pretrained ones. Consider Figure 3a, for example. The test task is to pick up a yellow box, but the agent has only learned how to pick up red objects, blue objects, yellow objects, and keys. It has not learned how to pick up boxes. However, we note from the inferred Boolean expression ($\widetilde{M}_1$) that the agent correctly identifies that the desired objects are, at the very least, yellow. Without further improvements to this transferred policy (SOPGOL-transfer), we can see that this approach outperforms DQN from the start. This is due to two main factors: (i) the transferred policy navigates to objects more reliably, so takes fewer random actions; and (ii) although the transferred policy does not have a complete understanding of which are the desirable objects, it at least navigates to yellow objects, which are sometimes yellow boxes.

Finally, since SOPGOL is able to determine that the current task is not entirely expressible in terms of its previous tasks (by checking whether $T_{SOP} = \tilde{T}$), it is able to learn a new $\tilde{Q}$-value function that improves on the transferred policy. Additionally, its returns are strictly higher than those of SOPGOL-transfer because SOPGOL learns the new $\bar{Q}$-value function faster by using $\bar{Q}_{SOP} \vee \tilde{Q}$ in the behaviour policy.

## 4.2 LIFELONG TRANSFER

In this section, we consider the more general setting where the agent is not necessarily given pretrained skills upfront, but is rather presented with tasks sampled from some unknown distribution. We revisit the example given in Section 3, but now more concretely by using a stochastic Four Rooms domain (Sutton et al., 1999), with a goal space of size $|\mathcal{G}| = 40$ and a task space of size $|\mathcal{M}| = 2^{|\mathcal{G}|} \approx 10^{12}$. Complete environment details are given in Appendix C.

We demonstrate the ability of SOPGOL to generalise over task distributions by evaluating the approach with the following distributions: (i) $\mathcal{D}_{sampled}$: the goals for each task are chosen uniformly at random over $\mathcal{G}$; (ii) $\mathcal{D}_{best}$: the first $\lceil \log_2 |\mathcal{G}| \rceil$ tasks are the base tasks, while the rest follow $\mathcal{D}_{sampled}$. This distribution gives the agent the minimum number of tasks to learn and store, since the agent learns the base tasks first before being presented with any other task. (iii) $\mathcal{D}_{worst}$: the first $|\mathcal{G}|$ tasks are each defined by a single goal that differs from the previous tasks, while the rest follow $\mathcal{D}_{sampled}$. This distribution forces the agent to learn and store the maximum number of tasks, since none of the $|\mathcal{G}|$ tasks can be expressed as a logical combination of the others. We use Q-learning (Watkins, 1989) as the RL method for SOPGOL, and Q-learning with $max\mathcal{Q}$ initialisation as a baseline. This has been shown by previous work (Abel et al., 2018) to be a practical method of initialising value functions with a theoretically optimal optimism criterion that speeds-up convergence during training. Our results (Figure 4) show that SOPGOL enables a lifelong agent to quickly generalise over an unknown task distribution. Interestingly, both graphs show that the convergence

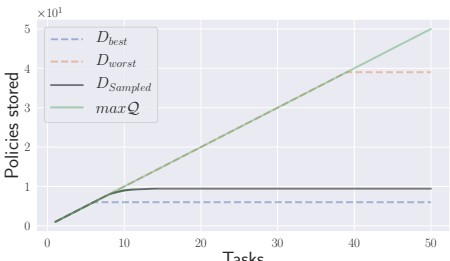

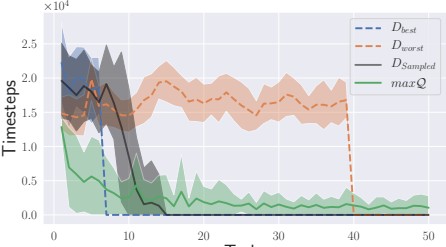

(a) Number of policies learned and stored after solving $n$ tasks.

(b) Number of samples required to learn $\epsilon$-optimal policies for each task.

Figure 4: Number of policies learned and samples required for the first 50 tasks of an agent's lifetime in the Four Rooms domain. The shaded regions represent standard deviations over 25 runs.

speed during a randomly sampled task distribution $\mathcal{D}_{sampled}$ is very close to that of the best task distribution $\mathcal{D}_{best}$. This suggests that there is room to make the bound in Theorem 2 even tighter by making some assumptions on the task distribution—an interesting avenue for future work.

## 5  RELATED WORK

There have been several approaches in recent years for tackling the problem of transfer in lifelong RL. Most closely related is the line of work on concurrent skill composition (Todorov, 2009; Saxe et al., 2017; Haarnoja et al., 2018; van Niekerk et al., 2019; Hunt et al., 2019). These methods usually focus on multi-goal tasks, where they address the combinatorial amount of desirable goals by composing learned skills to create new ones. Given a reward function that is well approximated by a linear function, Barreto et al. (2020) propose a scheme for few-shot transfer in RL by combining GPI and successor features (SF) (Barreto et al., 2017). In general, approaches based on GPI with SFs (Barreto et al., 2021) are suitable for tasks defined by linear preferences over features (latent goal states). Given the set of features for an environment, Alver & Precup (2022) shows that a base set of successor features can be learned, which is sufficient to span the task space. While these approaches also support tasks where goals are not terminal, the smallest number of successor features that must be learned to span the task space is $|\mathcal{G}|$ (the upper-bound in Theorem 2). Our work is similar to these approaches in that it can be interpreted as performing GPI with the logical composition of extended value functions, which leads to stronger theoretical bounds than GPI with the linear composition of successor features (see Appendix A.4). Finally, none of these works consider the lifelong RL setting where an agent starts with no skill and receives tasks sampled from an unknown distribution (without additional knowledge like base features or true task representations). In contrast, SOPGOL is able to handle this setting with logarithmic bounds on the number of skills needed to generalise over the task distribution (Theorem 2).

Other approaches like options (Sutton et al., 1999) and hierarchical RL (Barto & Mahadevan, 2003) address the lifelong RL problem via temporal compositions. These methods are usually focused on single-goal tasks, where they address the potentially long trajectories needed to reach a desired goal by composing sub-goal skills sequentially (Levy et al., 2017; Bagaria & Konidaris, 2019). While they do not consider the multi-goal setting, they can be used in conjunction with concurrent composition to learn how to achieve a combinatorial amount of desirable long horizon goals. Finally, there are also non-compositional approaches (Finn et al., 2017; Abel et al., 2018; Singh et al., 2021), which usually aim to learn the policy for a new task faster by initializing the networks with some pre-training procedure. These can be used in combination with SOPGOL to learn new skills faster.

## 6  CONCLUSION

In this work, we proposed an approach for efficient transfer learning in RL. Our framework, SOPGOL, leverages the Boolean algebra framework of Nangue Tasse et al. (2020) to determine which skills should be reused in a new task. We demonstrated that, if a new task is solvable using existing skills, an agent is able to solve it with no further learning. However, even if this is not the case, an estimate of the optimal value function can still be obtained to speed up training. This allows agents in a lifelong learning setting to quickly generalise over any unknown (possibly non-stationary) task distribution.

The main limitation of this work is that it only consider tasks with binary goal rewards—where goals are either desirable or not. Although this covers a vast number of many-goal tasks, combining our framework with works on weighted composition (van Niekerk et al., 2019; Barreto et al., 2020) could enable a similar level of generalisation over tasks with arbitrary goal rewards. Another exciting avenue for future work would be to extend our transfer learning and generalisation results to include temporal tasks by leveraging temporal composition approaches like options. Finally, we note that just like previous work, we rely on the existence of an off-the-shelf RL method that is able to learn goal-reaching tasks in a given environment. Since that is traditionally very sample inefficient, our framework can be complemented with other transfer learning methods like MAXQINIT (Abel et al., 2018) to speed up the learning of new skills (over and above the transfer learning and task space generalisation shown here). Our approach is a step towards the goal of truly general, long-lived agents, which are able to generalise both within tasks, as well as over the distribution of possible tasks it may encounter.

ACKNOWLEDGEMENTS

GNT is supported by an IBM PhD Fellowship. This research was supported, in part, by the National Research Foundation (NRF) of South Africa under grant number 117808. The content is solely the responsibility of the authors and does not necessarily represent the official views of the NRF.

The authors acknowledge the Centre for High Performance Computing (CHPC), South Africa, for providing computational resources to this research project. Computations were also performed using High Performance Computing Infrastructure provided by the Mathematical Sciences Support unit at the University of the Witwatersrand.

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

## A  PROOFS OF THEORETICAL RESULTS

### A.1  BOOLEAN ALGEBRA DEFINITION

**Definition 6.** *A Boolean algebra is a set $\mathcal{B}$ equipped with the binary operators $\vee$ (disjunction) and $\wedge$ (conjunction), and the unary operator $\neg$ (negation), which satisfies the following Boolean algebra axioms for $a, b, c$ in $\mathcal{B}$:*

*(i) Idempotence: $a \wedge a = a \vee a = a$.*

*(ii) Commutativity: $a \wedge b = b \wedge a$ and $a \vee b = b \vee a$.*

*(iii) Associativity: $a \wedge (b \wedge c) = (a \wedge b) \wedge c$ and $a \wedge (b \vee c) = (a \vee b) \vee c$.*

*(iv) Absorption: $a \wedge (a \vee b) = a \vee (a \wedge b) = a$.*

*(v) Distributivity: $a \wedge (b \vee c) = (a \wedge b) \vee (a \wedge c)$ and $a \vee (b \wedge c) = (a \vee b) \wedge (a \vee c)$.*

*(vi) Identity: there exists $\mathbf{0}, \mathbf{1}$ in $\mathcal{B}$ such that*

$$\mathbf{0} \wedge a = \mathbf{0}$$
$$\mathbf{0} \vee a = a$$
$$\mathbf{1} \wedge a = a$$
$$\mathbf{1} \vee a = \mathbf{1}$$

*(vii) Complements: for every $a$ in $\mathcal{B}$, there exists an element $a'$ in $\mathcal{B}$ such that $a \wedge a' = \mathbf{0}$ and $a \vee a' = \mathbf{1}$.*

### A.2  PROOFS FOR PROPOSITION 2

**Lemma 1.** *Let $\mathcal{M}$ be a set of tasks. Then $(\mathcal{M}, \vee, \wedge, \neg, \mathcal{M}_{MAX}, \mathcal{M}_{MIN})$ is a Boolean algebra.*

*Proof.* Let $M_1, M_2 \in \mathcal{M}$. We show that $\neg, \vee, \wedge$ satisfy the Boolean properties (i) – (vii).

**(i)–(v):** These easily follow from the fact that the $\min$ and $\max$ functions satisfy the idempotent, commutative, associative, absorption and distributive laws.

**(vi):** Let $r_{\mathcal{M}_{MAX} \wedge M_1}$ and $r_{M_1}$ be the reward functions for $\mathcal{M}_{MAX} \wedge M_1$ and $M_1$ respectively. Then for all $(s, a)$ in $\mathcal{S} \times \mathcal{A}$,

$$
r_{\mathcal{M}_{MAX} \wedge M_1}(s, a) = \begin{cases} \min\{r_{\text{MAX}}, r_{M_1}(s, a)\}, & \text{if } s \in \mathcal{G} \\ \min\{r_0(s, a), r_0(s, a)\}, & \text{otherwise.} \end{cases}
$$

$$
= \begin{cases} r_{M_1}(s, a), & \text{if } s \in \mathcal{G} \\ r_0(s, a), & \text{otherwise.} \end{cases} \qquad (r_{M_1}(s, a) \in \{r_{\text{MIN}}, r_{\text{MAX}}\} \text{ for } s \in \mathcal{G})
$$

$$
= r_{M_1}(s, a).
$$

Thus $\mathcal{M}_{MAX} \wedge M_1 = M_1$. Similarly $\mathcal{M}_{MAX} \vee M_1 = \mathcal{M}_{MAX}$, $\mathcal{M}_{MIN} \wedge M_1 = \mathcal{M}_{MIN}$, and $\mathcal{M}_{MIN} \vee M_1 = M_1$. Hence $\mathcal{M}_{MIN}$ and $\mathcal{M}_{MAX}$ are the universal bounds of $\mathcal{M}$.

**(vii):** Let $r_{M_1 \wedge \neg M_1}$ be the reward function for $M_1 \wedge \neg M_1$. Then for all $(s, a)$ in $\mathcal{S} \times \mathcal{A}$,

$$
r_{M_1 \wedge \neg M_1}(s, a) = \begin{cases} \min\{r_{M_1}(s, a), (r_{\text{MAX}} + r_{\text{MIN}}) - r_{M_1}(s, a)\}, & \text{if } s \in \mathcal{G} \\ \min\{r_0(s, a), (r_0(s, a) + r_0(s, a)) - r_0(s, a)\}, & \text{otherwise.} \end{cases}
$$

$$
= \begin{cases} r_{\text{MIN}}, & \text{if } s \in \mathcal{G} \text{ and } r_{M_1}(s, a) = r_{\text{MAX}} \\ r_{\text{MAX}}, & \text{if } s \in \mathcal{G} \text{ and } r_{M_1}(s, a) = r_{\text{MIN}} \\ r_0(s, a), & \text{otherwise.} \end{cases}
$$

$$
= r_{\mathcal{M}_{MIN}}(s, a).
$$

Thus $M_1 \wedge \neg M_1 = \mathcal{M}_{MIN}$, and similarly $M_1 \vee \neg M_1 = \mathcal{M}_{MAX}$.

$\square$

**Lemma 2.** *Let $\bar{\mathcal{Q}}^*$ be the set of optimal $\bar{Q}$-value functions for tasks in $\mathcal{M}$. Then $(\bar{\mathcal{Q}}^*, \vee, \wedge, \neg, \bar{Q}^*_{MAX}, \bar{Q}^*_{MIN})$ is a Boolean Algebra.*

*Proof.* Let $\bar{Q}^*_{M_1}, \bar{Q}^*_{M_2} \in \bar{\mathcal{Q}}^*$ be the optimal $\bar{Q}$-value functions for tasks $M_1, M_2 \in \mathcal{M}$ with reward functions $r_{M_1}$ and $r_{M_2}$. We show that $\neg, \vee, \wedge$ satisfy the Boolean properties (i) – (vii).

**(i)–(v):** These follow directly from the properties of the $\min$ and $\max$ functions.

**(vi):** For all $(s, g, a)$ in $\mathcal{S} \times \mathcal{G} \times \mathcal{A}$,

$$
\begin{aligned}
(\bar{Q}^*_{MAX} \wedge \bar{Q}^*_{M_1})(s, g, a) &= \min\{\bar{Q}^*_{MAX}(s, g, a), \bar{Q}^*_{M_1}(s, g, a)\} \\
&= \begin{cases} \min\{\bar{Q}^*_{MAX}(s, g, a), \bar{Q}^*_{MAX}(s, g, a)\}, & \text{if } r_{M_1}(g, a') = r_{\text{MAX}} \; \forall a' \in \mathcal{A} \\ \min\{\bar{Q}^*_{MAX}(s, g, a), \bar{Q}^*_{MIN}(s, g, a)\}, & \text{otherwise.} \end{cases} \\
&= \begin{cases} \bar{Q}^*_{MAX}(s, g, a), & \text{if } r_{M_1}(g, a) = r_{\text{MAX}} \; \forall a' \in \mathcal{A} \\ \bar{Q}^*_{MIN}(s, g, a), & \text{otherwise.} \end{cases} \\
&= \bar{Q}^*_{M_1}(s, g, a) \quad (\text{since } r_{M_1}(g, a') \in \{r_{\text{MIN}}, r_{\text{MAX}}\} \; \forall a' \in \mathcal{A}).
\end{aligned}
$$

Similarly, $\bar{Q}^*_{MAX} \vee \bar{Q}^*_{M_1} = \bar{Q}^*_{MAX}, \bar{Q}^*_{MIN} \wedge \bar{Q}^*_{M_1} = \bar{Q}^*_{MIN}$, and $\bar{Q}^*_{MIN} \vee \bar{Q}^*_{M_1} = \bar{Q}^*_{M_1}$.

**(vii):** For all $(.)$ in $\mathcal{S} \times \mathcal{G} \times \mathcal{A}$,

$$
\begin{aligned}
(\bar{Q}^*_{M_1} \wedge \neg\bar{Q}^*_{M_1})(.) &= \min\{\bar{Q}^*_{M_1}(.), \neg\bar{Q}^*_{M_1}(.)\} \\
&= \begin{cases} \min\{\bar{Q}^*_{MIN}(.), \bar{Q}^*_{MAX}(.)\} & \text{if } |\bar{Q}^*(.) - \bar{Q}^*_{MIN}(.)| \leq |\bar{Q}^*(.) - \bar{Q}^*_{MAX}(.)| \\ \min\{\bar{Q}^*_{MAX}(.), \bar{Q}^*_{MIN}(.)\} & \text{otherwise,} \end{cases} \\
&= \bar{Q}^*_{MIN}(.).
\end{aligned}
$$

Similarly, $\bar{Q}^*_{M_1} \vee \neg\bar{Q}^*_{M_1} = \bar{Q}^*_{MAX}$.

$\square$

**Lemma 3.** *Let $\bar{\mathcal{Q}}^*$ be the set of optimal extended $\bar{Q}$-value functions for tasks in $\mathcal{M}$. Then for all $M_1, M_2 \in \mathcal{M}$, we have (i) $\bar{Q}^*_{\neg M_1} = \neg\bar{Q}^*_{M_1}$, (ii) $\bar{Q}^*_{M_1 \vee M_2} = \bar{Q}^*_{M_1} \vee \bar{Q}^*_{M_2}$, and (iii) $\bar{Q}^*_{M_1 \wedge M_2} = \bar{Q}^*_{M_1} \wedge \bar{Q}^*_{M_2}$.*

*Proof.* Let $M_1, M_2 \in \mathcal{M}$. Then for all $(s, g, a)$ in $\mathcal{S} \times \mathcal{G} \times \mathcal{A}$,

**(i):**
$$
\begin{aligned}
&\bar{Q}^*_{\neg M_1}(s, g, a) \\
&= \begin{cases} \bar{Q}^*_{MAX}(s, g, a), & \text{if } r_{\neg M_1}(g, a') = r_{\text{MAX}} \; \forall a' \in \mathcal{A} \\ \bar{Q}^*_{MIN}(s, g, a), & \text{otherwise.} \end{cases} \\
&= \begin{cases} \bar{Q}^*_{MAX}(s, g, a), & \text{if } r_{M_1}(g, a') = r_{\text{MIN}} \; \forall a' \in \mathcal{A} \\ \bar{Q}^*_{MIN}(s, g, a), & \text{otherwise.} \end{cases} \\
&= \begin{cases} \bar{Q}^*_{MAX}(s, g, a), & \text{if } \bar{Q}^*_{M_1}(s, g, a) = \bar{Q}^*_{MIN}(s, g, a) \\ \bar{Q}^*_{MIN}(s, g, a), & \text{otherwise.} \end{cases} \\
&= \begin{cases} \bar{Q}^*_{MAX}(s, g, a), & \text{if } |\bar{Q}^*_{M_1}(s, g, a) - \bar{Q}^*_{MIN}(s, g, a)| \leq |\bar{Q}^*_{M_1}(s, g, a) - \bar{Q}^*_{MAX}(s, g, a)| \\ \bar{Q}^*_{MIN}(s, g, a), & \text{otherwise.} \end{cases} \\
&= \neg\bar{Q}^*_{M_1}(s, g, a).
\end{aligned}
$$

**(ii):**

$$\bar{Q}^*_{M_1 \vee M_2}(s,g,a) = \begin{cases} \bar{Q}^*_{MAX}(s,g,a), & \text{if } r_{M_1 \vee M_2}(g,a') = r_{\text{MAX}} \ \forall a' \in \mathcal{A} \\ \bar{Q}^*_{MIN}(s,g,a), & \text{otherwise.} \end{cases}$$

$$= \begin{cases} \bar{Q}^*_{MAX}(s,g,a), & \text{if } \max\{r_{M_1}(g,a'), r_{M_2}(g,a')\} = r_{\text{MAX}} \ \forall a' \in \mathcal{A} \\ \bar{Q}^*_{MIN}(s,g,a), & \text{otherwise.} \end{cases}$$

$$= \begin{cases} \bar{Q}^*_{MAX}(s,g,a), & \text{if } \max\{\bar{Q}^*_{M_1}(s,g,a), \bar{Q}^*_{M_2}(s,g,a)\} = \bar{Q}^*_{MAX}(s,g,a) \\ \bar{Q}^*_{MIN}(s,g,a), & \text{otherwise.} \end{cases}$$

$$= \max\{\bar{Q}^*_{M_1}(s,g,a), \bar{Q}^*_{M_2}(s,g,a)\}$$

$$= (\bar{Q}^*_{M_1} \vee \bar{Q}^*_{M_2})(s,g,a).$$

**(iii):** Follows similarly to (ii).

$\square$

**Proposition 2.** *Let $\bar{\mathcal{Q}}^*$ be the set of optimal $\bar{Q}$-value functions for tasks in $\mathcal{M}$. Let $\mathscr{A} : \mathcal{M} \to \bar{\mathcal{Q}}^*$ be any map from $\mathcal{M}$ to $\bar{\mathcal{Q}}^*$ such that $\mathscr{A}(M) = \bar{Q}^*_M$ for all $M$ in $\mathcal{M}$. Then,*

*(i) $\mathcal{M}$ and $\bar{\mathcal{Q}}^*$ respectively form a Boolean task algebra $(\mathcal{M}, \vee, \wedge, \neg, \mathcal{M}_{MAX}, \mathcal{M}_{MIN})$ and a Boolean extended value functions algebra $(\bar{\mathcal{Q}}^*, \vee, \wedge, \neg, \bar{Q}^*_{MAX}, \bar{Q}^*_{MIN})$,*

*(ii) $\mathscr{A}$ is a homomorphism between $\mathcal{M}$ and $\bar{\mathcal{Q}}^*$.*

*Proof.* **(i):** Follows from Lemma 1 and 2.

**(ii):** Follows from Lemma 3.

$\square$

### A.3 PROOFS FOR THEOREM 1

**Lemma 4.** *Let $\bar{\mathcal{Q}}^*$ be the set of optimal $\bar{Q}$-value functions for tasks in $\mathcal{M}$. Denote $\tilde{\bar{Q}}^*_M$ as the $\epsilon$-optimal $\bar{Q}$-value function for a task $M \in \mathcal{M}$ such that*

$$|\bar{Q}^*_M(s,g,a) - \tilde{\bar{Q}}^*_M(s,g,a)| \leq \epsilon \text{ for all } (s,g,a) \in \mathcal{S} \times \mathcal{G} \times \mathcal{A}.$$

*Then for all $M_1, M_2$ in $\mathcal{M}$ and $(s,g,a)$ in $\mathcal{S} \times \mathcal{G} \times \mathcal{A}$,*

*(i) $\left| [\bar{Q}^*_{M_1} \vee \bar{Q}^*_{M_2}](s,g,a) - [\tilde{\bar{Q}}^*_{M_1} \vee \tilde{\bar{Q}}^*_{M_2}](s,g,a) \right| \leq \epsilon$*

*(ii) $\left| [\bar{Q}^*_{M_1} \wedge \bar{Q}^*_{M_2}](s,g,a) - [\tilde{\bar{Q}}^*_{M_1} \wedge \tilde{\bar{Q}}^*_{M_2}](s,g,a) \right| \leq \epsilon$*

*(iii) $\left| \neg \bar{Q}^*_{M_1}(s,g,a) - \neg \tilde{\bar{Q}}^*_{M_1}(s,g,a) \right| \leq \epsilon$*

*Proof.* **(i):**

$$\left| [\bar{Q}^*_{M_1} \vee \bar{Q}^*_{M_2}](s,g,a) - [\tilde{\bar{Q}}^*_{M_1} \vee \tilde{\bar{Q}}^*_{M_2}](s,g,a) \right|$$

$$= \left| \max_{M \in \{M_1, M_2\}} \bar{Q}^*_M(s,g,a) - \max_{M \in \{M_1, M_2\}} \tilde{\bar{Q}}^*_M(s,g,a) \right|$$

$$\leq \max_{M \in \{M_1, M_2\}} \left| \bar{Q}^*_M(s,g,a) - \tilde{\bar{Q}}^*_M(s,g,a) \right|$$

$$\leq \epsilon.$$

**(ii):**

$$\left|[\bar{Q}^*_{M_1} \wedge \bar{Q}^*_{M_2}](s,g,a) - [\tilde{\bar{Q}}^*_{M_1} \wedge \tilde{\bar{Q}}^*_{M_2}](s,g,a)\right|$$

$$= \left|\min_{M \in \{M_1, M_2\}} \bar{Q}^*_M(s,g,a) - \min_{M \in \{M_1, M_2\}} \tilde{\bar{Q}}^*_M(s,g,a)\right|$$

$$\leq \min_{M \in \{M_1, M_2\}} \left|\bar{Q}^*_M(s,g,a) - \tilde{\bar{Q}}^*_M(s,g,a)\right|$$

$$\leq \epsilon.$$

**(iii):**

$$\left|\neg\bar{Q}^*_{M_1}(s,g,a) - \neg\tilde{\bar{Q}}^*_{M_1}(s,g,a)\right|$$

$$= \begin{cases} |\bar{Q}^*_{MAX}(s,g,a) - \neg\tilde{\bar{Q}}^*_{MIN}(s,g,a)|, & \text{if } \bar{Q}^*_{M_1} = \bar{Q}^*_{MIN}(s,g,a) \\ |\bar{Q}^*_{MIN}(s,g,a) - \neg\tilde{\bar{Q}}^*_{MAX}(s,g,a)|, & \text{otherwise.} \end{cases}$$

$$= \begin{cases} |\bar{Q}^*_{MAX}(s,g,a) - \tilde{\bar{Q}}^*_{MAX}(s,g,a)|, & \text{if } \bar{Q}^*_{M_1} = \bar{Q}^*_{MIN}(s,g,a) \\ |\bar{Q}^*_{MIN}(s,g,a) - \tilde{\bar{Q}}^*_{MIN}(s,g,a)|, & \text{otherwise.} \end{cases}$$

$$\leq \epsilon.$$

$\square$

**Lemma 5.** *Let $M \in \mathcal{M}$ be a task with reward function $r$, binary representation $T$ and optimal extended action-value function $\bar{Q}^*$. Given $\epsilon$-approximations of the binary representations $\tilde{\mathcal{T}}_n = \{\tilde{T}_1, ..., \tilde{T}_n\}$ and optimal $\bar{Q}$-functions $\tilde{\bar{\mathcal{Q}}}^*_n = \{\tilde{\bar{Q}}^*_1, ..., \tilde{\bar{Q}}^*_n\}$ for n tasks $\hat{\mathcal{M}} = \{M_1, ..., M_n\} \subseteq \mathcal{M}$, let*

$$T_\mathcal{F} = \mathcal{B}_{EXP}(\tilde{\mathcal{T}}_n) \text{ and } \bar{Q}_\mathcal{F} = \mathcal{B}_{EXP}(\tilde{\bar{\mathcal{Q}}}^*_n),$$

*where $\mathcal{B}_{EXP}$ is derived from $\tilde{\mathcal{T}}_n$ and $\tilde{T}$ using a generic method $\mathcal{F}$. Define $\pi(s) \in \arg\max_{a \in \mathcal{A}} Q_\mathcal{F}$ where $Q_\mathcal{F} := \max_{g \in \mathcal{G}} \bar{Q}_\mathcal{F}(s,g,a)$. Then,*

$$\|\bar{Q}^* - \bar{Q}_\mathcal{F}\|_\infty \leq (\mathbf{1}_{T \neq T_\mathcal{F}})r_\Delta + \epsilon,$$

*where $\mathbf{1}$ is the indicator function, $r_\Delta := r_{MAX} - r_{MIN}$, and $\|f - h\|_\infty := \max_{s,g,a} |f(s,g,a) - h(s,g,a)|$.*

*Proof.*
$$|\bar{Q}^*(s,g,a) - \bar{Q}_\mathcal{F}(s,g,a)| = |\bar{Q}^*(s,g,a) - \bar{Q}^*_\mathcal{F}(s,g,a) + \bar{Q}^*_\mathcal{F}(s,g,a) - \bar{Q}_\mathcal{F}(s,g,a)|$$
$$\leq |\bar{Q}^*(s,g,a) - \bar{Q}^*_\mathcal{F}(s,g,a)| + |\bar{Q}^*_\mathcal{F}(s,g,a) - \bar{Q}_\mathcal{F}(s,g,a)|$$
$$\leq |\bar{Q}^*(s,g,a) - \bar{Q}^*_\mathcal{F}(s,g,a)| + \epsilon. \qquad \text{(Using Lemma 4)}$$

If $T = T_\mathcal{F}$, then $\bar{Q}^*(s,g,a) = \bar{Q}^*_\mathcal{F}(s,g,a)$, and we are done. Let $T \neq T_\mathcal{F}$. Without loss of generality, let $\bar{Q}^*(s,g,a) = \bar{Q}^*_{MAX}(s,g,a)$ and $\bar{Q}^*_\mathcal{F}(s,g,a) = \bar{Q}^*_{MIN}(s,g,a)$. Then,

$$|\bar{Q}^*(s,g,a) - \bar{Q}^*_\mathcal{F}(s,g,a)| \leq |\bar{Q}^*_{MAX}(s,g,a) - \bar{Q}^*_{MIN}(s,g,a)|$$
$$\leq r_\Delta.$$

$\square$

**Lemma 6.** *Let $Q^*$ and $\bar{Q}^*$ be the optimal Q-value function and optimal extended Q-value function respectively for a deterministic task in $\mathcal{M}$. Then for all $(s,a)$ in $\mathcal{S} \times \mathcal{A}$, we have*

$$Q^*(s,a) = \max_{g \in \mathcal{G}} \bar{Q}^*(s,g,a).$$

*Proof.* We first note that

$$\max_{g \in \mathcal{G}} \bar{r}(s, g, a) = \begin{cases} \max\{r_{\mathsf{MIN}}, r(s,a)\}, & \text{if } s \in \mathcal{G} \\ \max_{g \in \mathcal{G}} r(s,a), & \text{otherwise.} \end{cases} = r(s,a). \tag{4}$$

Now define

$$\bar{Q}^*_{max}(s,a) := \max_{g \in \mathcal{G}} \bar{Q}^*(s,g,a).$$

Then it follows that

$$\left[\mathcal{T}\bar{Q}^*_{max}\right](s,a) = r(s,a) + \gamma \sum_{s' \in \mathcal{S}} p(s'|s,a) \max_{a' \in \mathcal{A}} \bar{Q}^*_{max}(s',a')$$

$$= r(s,a) + \gamma \sum_{s' \in \mathcal{S}} p(s'|s,a) \max_{a' \in \mathcal{A}} \left[\max_{g \in \mathcal{G}} \bar{Q}^*(s',g,a')\right]$$

$$= r(s,a) + \gamma \sum_{s' \in \mathcal{S}} p(s'|s,a) \max_{g \in \mathcal{G}} \left[\max_{a' \in \mathcal{A}} \bar{Q}^*(s',g,a')\right]$$

$$= r(s,a) + \max_{g \in \mathcal{G}} \left[\gamma \sum_{s' \in \mathcal{S}} p(s'|s,a) \max_{a' \in \mathcal{A}} \bar{Q}^*(s',g,a')\right] \quad \text{(Since } p \text{ is deterministic)}$$

$$= \max_{g \in \mathcal{G}} \bar{r}(s,g,a) + \max_{g \in \mathcal{G}} \left[\gamma \sum_{s' \in \mathcal{S}} p(s'|s,a) \max_{a' \in \mathcal{A}} \bar{Q}^*(s',g,a')\right] \quad \text{(Using Equation 4)}$$

$$= \max_{g \in \mathcal{G}} \left[\bar{r}(s,g,a) + \gamma \sum_{s' \in \mathcal{S}} p(s'|s,a) \max_{a' \in \mathcal{A}} \bar{Q}^*(s',g,a')\right],$$

since $\bar{r}(s,g,a) = r_0(s,a) \,\forall s \notin \mathcal{G}$ and $p(s,a,\omega) = 1$ with $\bar{Q}^*(\omega,g,a') = 0 \,\forall s \in \mathcal{G}$.

$$= \max_{g \in \mathcal{G}} \bar{Q}^*(s,g,a)$$

$$= \bar{Q}^*_{max}(s,a).$$

Hence $\bar{Q}^*_{max}$ is a fixed point of the Bellman optimality operator.

If $s \in \mathcal{G}$, then

$$\bar{Q}^*_{max}(s,a) = \max_{g \in \mathcal{G}} Q^*(s,g,a) = \max_{g \in \mathcal{G}} \bar{r}(s,g,a) = r(s,a) = Q^*(s,a).$$

Since $\bar{Q}^*_{max} = Q^*$ holds in $\mathcal{G}$ and $\bar{Q}^*_{max}$ is a fixed point of the Bellman operator, then $\bar{Q}^*_{max} = Q^*$ holds everywhere.

$\square$

**Theorem 1.** *Let $M \in \mathcal{M}$ be a task with reward function $r$, binary representation $T$ and optimal extended action-value function $\bar{Q}^*$. Given $\epsilon$-approximations of the binary representations $\tilde{\mathcal{T}}_n = \{\tilde{T}_1, ..., \tilde{T}_n\}$ and optimal $\bar{Q}$-functions $\tilde{\bar{\mathcal{Q}}}^*_n = \{\tilde{\bar{Q}}^*_1, ..., \tilde{\bar{Q}}^*_n\}$ for n tasks $\hat{\mathcal{M}} = \{M_1, ..., M_n\} \subseteq \mathcal{M}$, let*

$$T_{\mathcal{F}} = \mathcal{B}_{EXP}(\tilde{\mathcal{T}}_n) \text{ and } \bar{Q}_{\mathcal{F}} = \mathcal{B}_{EXP}(\tilde{\bar{\mathcal{Q}}}^*_n),$$

*where $\mathcal{B}_{EXP}$ is derived from $\tilde{\mathcal{T}}_n$ and $\tilde{T}$ using a generic method $\mathcal{F}$. Define $\pi(s) \in \arg\max_{a \in \mathcal{A}} Q_{\mathcal{F}}$ where $Q_{\mathcal{F}} := \max_{g \in \mathcal{G}} \bar{Q}_{\mathcal{F}}(s, g, a)$. Then,*

*(i)* $\|Q^* - Q^\pi\|_\infty \le \frac{2}{1-\gamma}((\mathbf{1}_{T \ne T_{\mathcal{F}}} + \mathbf{1}_{r \notin \{r_g\}_{|\mathcal{G}|}})r_\Delta + \epsilon),$

*(ii) if the dynamics are deterministic,*

$$\|Q^* - Q_{\mathcal{F}}\|_\infty \le (\mathbf{1}_{T \ne T_{\mathcal{F}}})r_\Delta + \epsilon,$$

where $\mathbf{1}$ is the indicator function, $r_g(s,a) := \bar{r}(s,g,a)$, $r_\Delta := r_{MAX} - r_{MIN}$, and $\|f-h\|_\infty :=$ $\max_{s,g,a} |f(s,g,a) - h(s,g,a)|$.

*Proof.* **(i):** We first note that each $g$ in $\mathcal{G}$ can be thought of as defining an MDP $M_g :=$ $(\mathcal{S}, \mathcal{A}, p, r_g, \gamma)$ with reward function $r_g(s,a) := \bar{r}(s,g,a)$, optimal policy $\pi_g^*(s) = \bar{\pi}^*(s,g)$ and optimal Q-value function $Q^{\pi_g^*}(s,a) = \bar{Q}^*(s,g,a)$. Then this proof follows similarly to that of Barreto et al. (2017) Theorem 2,

$Q^*(s,a) - Q^\pi(s,a)$

$\leq Q^*(s,a) - Q^{\pi_g^*}(s,a) + \frac{2}{1-\gamma}((\mathbf{1}_{T \neq T_\mathcal{F}})r_\Delta + \epsilon)$   (Barreto et al. (2017) Theorem 1)

$\leq \frac{2}{1-\gamma} \max_{s,a} |r(s,a) - r_g(s,a)| + \frac{2}{1-\gamma}((\mathbf{1}_{T \neq T_\mathcal{F}})r_\Delta + \epsilon)$   (Barreto et al. (2017) Lemma 1)

$\leq \frac{2}{1-\gamma}(\mathbf{1}_{r \neq r_g})r_\Delta + \frac{2}{1-\gamma}((\mathbf{1}_{T \neq T_\mathcal{F}})r_\Delta + \epsilon)$

(Since rewards only differ in $\mathcal{G}$ where $r(s,a), r_g(s,a) \in \{r_{MIN}, r_{MAX}\}$ for $s \in \mathcal{G}$)

$\leq \frac{2}{1-\gamma}((\mathbf{1}_{T \neq T_\mathcal{F}} + \mathbf{1}_{r \neq r_g})r_\Delta + \epsilon)$.

Hence,

$$\|Q^* - Q^\pi\|_\infty \leq \frac{2}{1-\gamma}((\mathbf{1}_{T \neq T_\mathcal{F}} + \min_g \mathbf{1}_{r \neq r_g})r_\Delta + \epsilon)$$

$$\leq \frac{2}{1-\gamma}((\mathbf{1}_{T \neq T_\mathcal{F}} + \mathbf{1}_{r \notin \{r_g\}_{|\mathcal{G}|}})r_\Delta + \epsilon)$$

$$\text{(Since } \min_g \mathbf{1}_{r \neq r_g} = 0 \text{ only when } r \in \{r_g\}_{|\mathcal{G}|} ).$$

**(ii):**

$$|Q^*(s,a) - Q_\mathcal{F}(s,a)| = |\max_g \bar{Q}^*(s,g,a) - \max_g \bar{Q}_\mathcal{F}(s,g,a)| \quad \text{(Lemma 6)}$$

$$\leq \max_g |\bar{Q}^*(s,g,a) - \bar{Q}_\mathcal{F}(s,g,a)|$$

$$\leq (\mathbf{1}_{T \neq T_\mathcal{F}})r_\Delta + \epsilon. \quad \text{(Lemma 5)}$$

$\square$

### A.4   COMPARING THE BOUNDS OF THEOREM 1 WITH THAT OF GPI IN BARRETO ET AL. (2018)

We first restate Proposition 1 (Barreto et al., 2018) here.

**Proposition 3** ((Barreto et al., 2018)). *Let $M \in \mathcal{M}$ and let $Q_i^{\pi_j^*}$ be the action value function of an optimal policy of $M_j \in \mathcal{M}$ when executed in $M_i \in \mathcal{M}$. Given approximations $\{\tilde{Q}_i^{\pi_1}, ..., \tilde{Q}_i^{\pi_n}\}$ such that $|Q_i^{\pi_j} - \tilde{Q}_i^{\pi_j}| \leq \epsilon$ for all $s,a \in \mathcal{S} \times \mathcal{A}$, and $j \in \{1, ..., n\}$, let*

$$\pi(s) \in \arg\max_a \max_j \tilde{Q}_i^{\pi_j}(s,a).$$

*then,*

$$\|Q^* - Q^\pi\|_\infty \leq \frac{2}{1-\gamma}(\|r - r_i\|_\infty + \min_j \|r_i - r_j\|_\infty + \epsilon),$$

*where $Q^*$ is the optimal value function of $M$, $Q^\pi$ is the value function of $\pi$ in $M$, and $\|f-h\|_\infty :=$ $\max_{s,g,a} |f(s,g,a) - h(s,g,a)|$.*

We can simplify the bound in Proposition 3 as follows:

$$\|Q^* - Q^\pi\|_\infty \le \frac{2}{1-\gamma}(\|r - r_i\|_\infty + \min_j \|r_i - r_j\|_\infty + \epsilon)$$

$$\le \frac{2}{1-\gamma}((\mathbf{1}_{r \ne r_i})r_\Delta + \min_j \|r_i - r_j\|_\infty + \epsilon)$$

(Since rewards only differ in $\mathcal{G}$ where $r(s,a), r_i(s,a) \in \{r_{\text{MIN}}, r_{\text{MAX}}\}$ for $s \in \mathcal{G}$)

$$\le \frac{2}{1-\gamma}((\mathbf{1}_{r \ne r_i})r_\Delta + (\min_j \mathbf{1}_{r_i \ne r_j})r_\Delta + \epsilon)$$

$$\le \frac{2}{1-\gamma}((\mathbf{1}_{r \ne r_i})r_\Delta + (\mathbf{1}_{r_i \notin \{r_j\}_n})r_\Delta + \epsilon)$$

(Since $\min_j \mathbf{1}_{r_i \ne r_j} = 0$ only when $r_i \in \{r_j\}_n$ )

$$\le \frac{2}{1-\gamma}((\mathbf{1}_{r \ne r_i} + \mathbf{1}_{r_i \notin \{r_j\}_n})r_\Delta + \epsilon).$$

where $\mathbf{1}$ is the indicator function, and $r_\Delta := r_{\text{MAX}} - r_{\text{MIN}}$. We can see that this bound is similar to that of Theorem 1(i) but weaker. This because:

(i) The first term of this bound ($\mathbf{1}_{r \ne r_i}$) requires that reward function of the current task ($r$) be identical to that of a reference task ($r_i$). In Barreto et al. (2018), $r_i$ is taken as the best linear approximation of $r$. In contrast, the first term of Theorem 1(i) ($\mathbf{1}_{T \ne T_{\mathcal{F}}}$) only requires the current task to be expressible as a Boolean composition of past tasks.

(ii) The second term of this bound ($\mathbf{1}_{r_i \notin \{r_j\}_n}$) requires that the reference task (the best linear approximation to the current task) is exactly one of the past tasks. In contrast, the second term of Theorem 1(i) ($\mathbf{1}_{r \notin \{r_g\}_{|\mathcal{G}|}}$) only requires the current task to have a single desirable goal.

This suggests that we can can think of the logical composition approach as an efficient way of doing GPI, one which leads to tight performance bounds on the transferred policy (Theorem 1(ii)).

## A.5 Proofs for Theorem 2

**Theorem 2.** *Let $\mathcal{D}$ be an unknown distribution, possibly non-stationary, over a set of tasks $\mathcal{M}(\mathcal{S}, \mathcal{A}, p, \gamma, r_0)$. Let $\mathscr{A} : \mathcal{M} \to \bar{\mathcal{Q}}^*$ be any map from $\mathcal{M}$ to $\bar{\mathcal{Q}}^*$ such that $\mathscr{A}(M) = \bar{Q}^*_M$ for all $M$ in $\mathcal{M}$. Let*

$$\tilde{\mathcal{T}}_{t+1}, \tilde{\bar{\mathcal{Q}}}^*_{t+1} = SOPGOL(\mathscr{A}, M_t, \tilde{\mathcal{T}}_t, \tilde{\bar{\mathcal{Q}}}^*_t) \text{ where } M_t \sim \mathcal{D}(t) \text{ and } \tilde{\mathcal{T}}_0 = \tilde{\bar{\mathcal{Q}}}^*_0 = \varnothing \ \forall t \in \mathbb{N}.$$

*Then,*

$$\lceil \log |\mathcal{G}| \rceil \le \lim_{t \to \infty} N_t \le |\mathcal{G}| \quad \text{where } N_t := |\tilde{\mathcal{T}}_t| = |\tilde{\bar{\mathcal{Q}}}^*_t|.$$

*Proof.* Let $\tilde{T}_t$ be the approximate binary representation of task $M_t$ learned by SOPGOL. We first note that SOPGOL returns $\tilde{\mathcal{T}}_t \cup \{\tilde{T}_t\}$ only if $\tilde{T}_t$ is not in the span of $\tilde{\mathcal{T}}_t$. That is,

$$\tilde{\mathcal{T}}_{t+1} = \tilde{\mathcal{T}}_t \cup \{\tilde{T}_t\} \text{ iff } \tilde{T}_t \ne \mathcal{B}_{EXP}(\tilde{\mathcal{T}}_t) \text{ where } \mathcal{B}_{EXP} = SOP(\tilde{\mathcal{T}}_t, \tilde{T}_t).$$

Hence, it is sufficient to show that the number, $N$, of linearly independent binary vectors, $\tilde{T} \in \{0,1\}^{|\mathcal{G}|}$, that span the Boolean vector space (Subrahmanyam, 1964), $GF(2)^{|\mathcal{G}|}$,[5] is bounded by

$$\lceil \log |\mathcal{G}| \rceil \le N \le |\mathcal{G}|.$$

This follows from the fact that $\lceil \log |\mathcal{G}| \rceil$ is the size of a minimal set of generators for $GF(2)^{|\mathcal{G}|}$ (as can easily be seen with a Boolean table), and $|\mathcal{G}|$ is its dimensionality.

$\square$

---

[5]GF(2) is the Galois field with two elements, $(\{0,1\}, +, .)$, where $+ := XOR$ and $. := AND$.

# B  SUM OF PRODUCTS WITH GOAL ORIENTED LEARNING

Algorithm 1 shows the full pseudo-code for SOPGOL. Here, $SOP(\tilde{\mathcal{T}}, \tilde{T})$ is the classical sum of products method in Boolean logic. Given a list of binary vectors $\mathcal{T} = [\tilde{T}_1, ..., \tilde{T}_n]$, a Boolean expression for a new binary vector $\tilde{T}$ is obtained as follows:

1. Identify all rows of $\tilde{T}$ with a 1.
2. For each such row: Make a product (conjunction) of all the input variables and make the negation of each variable with a 0 in this row.
3. Take the sum (disjunction) of all these product terms.

The output of $SOP$ is a function $B_{EXP}$ that takes in $|\tilde{\mathcal{T}}|$ variables, and applies disjunctions, conjunctions and negations to them according to the Boolean expression obtained above.

---

**Algorithm 1:** SOPGOL

---

**Input :** off-policy RL algorithm $\mathscr{A}$,                    /* e.g DQN */
         task MDP $M$,
         set of $\epsilon$-optimal task binary representations $\tilde{\mathcal{T}}$,
         set of $\epsilon$-optimal $\bar{Q}$-value functions $\tilde{\mathcal{Q}}$.

Initialise $\tilde{T} : \mathcal{G} \rightarrow \{0,1\}$

Initialise $\tilde{\tilde{Q}} : \mathcal{S} \times \mathcal{G} \times \mathcal{A} \rightarrow \mathbb{R}$ according to $\mathscr{A}$

Initialise goal buffer $\tilde{\mathcal{G}}$ with terminal states observed from a random policy

**while** $\tilde{\tilde{Q}}$ *is not converged* **do**
    Initialise state $s$ from $M$
    $\mathcal{B}_{EXP} \leftarrow SOP(\tilde{\mathcal{T}}, \tilde{T})$
    $T_{SOP}, \bar{Q}_{SOP} \leftarrow \mathcal{B}_{EXP}(\tilde{\mathcal{T}}), \mathcal{B}_{EXP}(\tilde{\mathcal{Q}}^*)$
    $\bar{Q} \leftarrow \bar{Q}_{SOP}$ **if** $\tilde{T} = T_{SOP}$ **else** $\tilde{\tilde{Q}} \vee \bar{Q}_{SOP}$
    $g \leftarrow \underset{g' \in \tilde{\mathcal{G}}}{\arg\max} \left( \underset{a \in \mathcal{A}}{\max} \bar{Q}(s, g', a) \right)$
    **while** $s$ *is not terminal* **do**
        Select action $a$ using the behaviour policy from $\mathscr{A}$: $a \leftarrow \bar{\pi}(s,g)$  /* e.g $\epsilon$-greedy */
        Take action $a$, observe reward $r$ and next state $s'$ in $M$
        **if** $\tilde{T} \neq T_{SOP}$ **then**
            **foreach** $g' \in \tilde{\mathcal{G}}$ **do**
                $\bar{r} \leftarrow r_{\text{MIN}}$ **if** $g' \neq s \in \tilde{\mathcal{G}}$ **else** $r$
                Update $\tilde{\tilde{Q}}$ with $(s, g', a, \bar{r}, s')$ according to $\mathscr{A}$
            **end**
        **if** $s$ *is terminal* **then**
            $\tilde{T}(s) \leftarrow \mathbf{1}_{r=r_{\text{MAX}}}$
            $\tilde{\mathcal{G}} \leftarrow \tilde{\mathcal{G}} \cup \{s\}$
        **else**
            $s \leftarrow s'$
        **end**
    **end**
**end**
$\mathcal{B}_{EXP} \leftarrow SOP(\tilde{\mathcal{T}}, \tilde{T})$
$\tilde{\mathcal{T}}, \tilde{\tilde{\mathcal{Q}}} \leftarrow (\tilde{\mathcal{T}}, \tilde{\tilde{\mathcal{Q}}})$ **if** $\tilde{T} = \mathcal{B}_{EXP}(\tilde{\mathcal{T}})$ **else** $(\tilde{\mathcal{T}} \cup \{\tilde{T}\}, \tilde{\tilde{\mathcal{Q}}} \cup \{\tilde{\tilde{Q}}\})$
**return** $\tilde{\mathcal{T}}, \tilde{\tilde{\mathcal{Q}}}$

---

## C    FOUR ROOMS ENVIRONMENT

We use the Four Rooms domain (Sutton et al., 1999), where an agent must navigate in a grid world to particular locations. The goal locations are placed along the sides of the walls and at the centre of rooms (Figure 5). This gives a goal space of size $|\mathcal{G}| = 40$ and a task space of size $|\mathcal{M}| = 2^{|\mathcal{G}|} \approx 10^{12}$. The agent can move in any of the four cardinal directions at each timestep, but colliding with a wall leaves the agent in the same location. We add a 5th action for "stay" that the agent chooses to achieve goals. A goal location only becomes terminal if the agent chooses to stay in it. All rewards are $0$ at non-terminal states, and $1$ at the desirable goals. The transition dynamics are stochastic with a slip probability ($sp = 0.1$). That is, with probability *1-sp* the agent moves in the direction it chooses, and with probability *sp* it moves in one of the other three chosen uniformly at random.

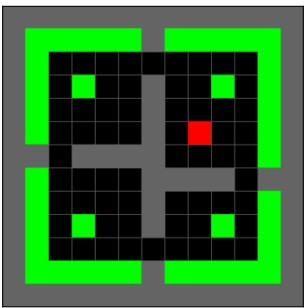

Figure 5: 40 goals Four Rooms domain with goals in green and the agent in red.

# D  FUNCTION APPROXIMATION EXPERIMENT DETAILS

## D.1  ENVIRONMENT

The PICKUPOBJ environment is fully observable, where each state observation is a $56 * 56 * 3$ RGB image (Figure 1). The agent has 7 actions it can take in this environment corresponding to: 1 - rotate left, 2 - rotate right, 3 - move one step forward if there is no wall or object in front, 4 - pickup object if there is an object in front and no object has been picked, 5 - drop the object in front if an object has been picked and there is no wall or object in front, 6 - open the door in front if there is a closed-door in front, and 7 - close the door in front if there is an opened door in front.

For each task, each episode starts with 1 desirable object and 4 other randomly chosen objects placed randomly in the environment. The agent is also placed at a random position with a random orientation at the start of each episode. The agent receives a reward of -0.1 at every timestep, and a reward of 2 when it picks up a desirable object. The environment transitions to a terminal state once the agent picks up any object and the agent observes the picked object. There are 15 types of objects (illustrated in Table 1) resulting in 15 possible goal states. Hence, the dimension of the state space is $|\mathcal{S}| = 56 \times 56 \times 3$, the goal space is $|\mathcal{G}| = 15$, and the action space is $|\mathcal{A}| = 7$.

## D.2  NETWORK ARCHITECTURE AND HYPERPARAMETERS

In our function approximation experiments, we represent each extended value function $\tilde{Q}^*$ with a list of $|\mathcal{G}|$ DQNs, such that the value function for each goal $\tilde{Q}_g^*(s, a) := \tilde{Q}^*(s, g, a)$ is approximated with a separate DQN. The DQNs used have the following architecture, with the CNN part being identical to that used by Mnih et al. (2015):

1. Three convolutional layers:
   (a) Layer 1 has 3 input channels, 32 output channels, a kernel size of 8 and a stride of 4.
   (b) Layer 2 has 32 input channels, 64 output channels, a kernel size of 4 and a stride of 2.
   (c) Layer 3 has 64 input channels, 64 output channels, a kernel size of 3 and a stride of 1.
2. Two fully-connected linear layers:
   (a) Layer 1 has input size 3136 and output size 512 and uses a ReLU activation function.
   (b) Layer 2 has input size 512 and output size 7 with no activation function.

We used the ADAM optimiser with batch size 256 and a learning rate of $10^{-3}$. We started training after 1000 steps of random exploration and updated the target Q-network every 1000 steps. Finally, we used $\epsilon$-greedy exploration, annealing $\epsilon$ from 0.5 to 0.05 over 100000 timesteps.

Finally, we used the same DQN architecture and training hyperparameters for the baseline in all experiments.

