# OpenReview forum: "Generalisation in Lifelong Reinforcement Learning through Logical Composition "
_ICLR.cc/2022/Conference — ICLR 2022 Poster_

### Official Review · Reviewer_KyZj · 2021-10-24

**Correctness:** 3
**Technical Novelty And Significance:** 3
**Empirical Novelty And Significance:** 3
**Recommendation:** 5
**Confidence:** 3

**Main Review:**

## Strengths

* This work extends the previous results of (Nangue Tasse et al. 2020) to handle stochastic transitions.
* The topic area is relevant for ICLR.
* Lifelong learning is an important and challenging topic and this work proposes a compelling path toward addressing that challenge through a combination of reinforcement learning and logical composition.
* Overall the paper is well-organized and does not have too many typos, although see my notes below.
* The theoretical results may be of interest, although I had difficulty understanding some of the main statements, which hopefully I can clear up during the rebuttal period.

## Weaknesses

* The paper is overall difficult to follow and the notation is not always clear.
* I had a lot of trouble parsing the statement of Theorem 1.
   * SOP(., .) is not defined, where the first argument is a set of binary vectors and the second argument is a single binary vector. The result of SOP is B_exp, which is then applied to \tilde{T}_n as if B_exp were a function (and similarly for Q)? I think I ultimately understand this part, but the notation seems abused to the point where it’s difficult to follow.
   * \tilde{T} (the approximation of T, presumably) is not defined within the scope of this theorem -- is it meant to appear in that line or not? I suspect that it was meant to be T, because the theorem does not have any assumptions about the accuracy of \tilde{T} as an approximation of T, and I’m sure that some assumptions would be required to draw any conclusions.
   * In (i), what is the meaning of $1_{T \not\in \tilde{T}_n}$? $T$ is a specification and $\tilde{T}_n$ is a set of specifications, so that subscript would either be true or false. But from context, this object must be a binary vector. What is it exactly?
   * Since I remain confused about the Theorem 1 formal statement, for now, I am trusting the written descriptions surrounding it.
* I also have some confusions surrounding Theorem 2:
   * Why are the cardinalities all that we care about? Is there an implication that a task and Q function are only included in those sets if they have been “mastered” with a certain level of accuracy?
   * How can we draw any conclusions when we have an arbitrary nonstationary distribution of tasks? What prevents a pathological example where we see the same task for the first T steps and then a different one on step T+1, for any T?
* As a delta on the previous work of (Nangue Tasse et al. 2020), I am not sure if this work constitutes a substantial enough standalone contribution for an ICLR paper. There are advances to be sure, but they are relatively incremental. If I am able to get a better understanding of the theorems, that may change my perspective.
* The experiments are good to illustrate the main ideas, but they are not a substantial standalone contribution, since the domains are very simple and the baseline comparisons are very limited.
* I was surprised by the new definition of Q-function negation in section 3.2 because the value of the Q-function before negation is not used, except when determining whether the output should “toggle” between Qmax or Qmin. Can you explain this?
* I would like to see a discussion of the relationship between this line of work and propositional AI planning. The composition of goals is very related. It will be easy to find differences, and I think this line of work may offer a different and valuable perspective. I just think that the work would be strengthened by deeply exploring these connections.
* The premise of the transfer results in this work is that the task specification for a given task is not known, and so we have to compute and use an approximation of it. I am having trouble imagining a scenario where this structured space of possible task specifications is known, but the specification for a given task is not known. Is the idea that for a given task, there may actually be no “true” task specification, and so the approximation is just the “closest” specification that we can find? That would be more compelling, but I don’t think that we can use this interpretation because it would require leaving the class of MDPs defined in this work, and including MDPs that do not necessarily have the good-goal/bad-goal structure with the fixed r0 reward function.
* This comment is about boolean task algebra for RL in general, not just the present work. Though the theoretical contributions here may be interesting in their own right, I am struggling to imagine a practical scenario where it would be preferable to use this framework instead of one where we learn or compute a universal goal-conditioned value function instead. To elaborate, here is my attempt to describe this line of work through the lens of universal goal-conditioned value function learning:
   * Say that we compute/learn a universal goal-conditioned Q function $Q(s, g, a)$. Given any goal state $g$, we can do $argmax_{a} Q(s, g, a)$ to get optimal actions.
   * This line of work says: instead of learning a single goal-conditioned Q function, we could compute/learn multiple goal-conditioned Q functions, one per “task”, where a task is a set of goal states. Each of these Q functions is like an “expert” about some part of the goal space.
   * When given a new task $\mathcal{G}$, if we had a universal $Q$, then we could recover an optimal action via $argmax_{a} \max_{g \in \mathcal{G}} Q(s, g, a)$. This would be “zero-shot” transfer in the sense of this paper.
   * But instead, if we only have the goal-space expert Q functions, we will query the experts for whom $\mathcal{G}$ is within their expertise and combine their outputs according to the relationships between $\mathcal{G}$ and the expertises. This too will accomplish “zero-shot” transfer. But what is gained over the universal approach?
* Related to the previous point, the super-exponential zero-shot generalization advertised in this work and the previous work strikes me as a little bit disingenuous. If I am able to do X, and I’m also able to do Y, then I am automatically also able to do “X or Y”. Is that really generalization? Fundamentally, that is my reading on what’s happening in this work -- we’re learning how to accomplish each of the goals (indirectly, through the tasks, which are sets of goals), and then we’re recombining them in different ways, so that if we know how to achieve goal 1 and goal 2, we also know how to achieve “goal 1 or goal 2.”


## Minor Points

* In the first sentence of the paragraph before Definition 3, is that & meant to be there? I think there may be a formatting error.
* I know that the notation $g \neq s \in \mathcal{G}$ is not original to this work, but it is confusing and should be changed. For example, two different ways to parse this: $\forall s \in \mathcal{G}. g \neq s$; or $\exists s \in \mathcal{G}. g \neq s$. Furthermore, $\mathcal{G} \subseteq \mathcal{S}$, so can’t we just say $g \not\in \mathcal{G}$?
* It would be useful to define SOP explicitly even if it’s simple.
* When first defining the approximate task specification $\widetilde{T}$, it would be helpful to say whether that object is also a binary vector, like $T$, or if it instead could have non-binary values. Since it’s an approximation and there is some uncertainty about the true specification, the latter is plausible.

**Summary Of The Paper:**

This paper considers lifelong reinforcement learning. It extends a line of recent work in which logical composition is combined with goal-based reinforcement learning to achieve generalization to new tasks. The main contribution is theoretical, extending prior work to handle stochastic transitions and establishing asymptotic results for the lifelong learning setting. Experiments in grid-based domains illustrate SOPGOL, the main proposed algorithm, and show that it is possible to immediately generalize to new tasks in the lifelong setting.

**Summary Of The Review:**

For the reasons described above, especially concerning clarity and novelty, I am recommending a weak reject at this stage. However, after getting a better understanding of the theoretical contributions, I may be willing to raise my score.

---

> ### Author Response · Authors · 2021-11-21
> **Reply to Reviewer KyZj [1/4]**
>
> Thank you for your careful review of our paper. We hope that the following points address your concerns.
>
>
> 1. > As a delta on the previous work of (Nangue Tasse et al. 2020), I am not sure if this work constitutes a substantial enough standalone contribution for an ICLR paper.
>    * In brief, our approach says the following: When an agent is presented with a new task, it is able to learn what task it is required to solve in a supervised way (by learning its binary specification from rewards). It is then able to generate a boolean expression to determine whether or not that task is solvable zero-shot using its acquired skills (learned Q-functions). If yes, then it is able to produce the Q-function that solves the current task without any further learning. If not, it is able speed up the RL of the current task by using its estimated Q-function (few-shot).
>    * To the best of our knowledge, this is the first work that achieves such a degree of transfer learning and generalization.
>       1. Tasse et al. 2020 achieved only zero-shot transfer (i.e no few-shot transfer) when the *optimal* value functions are learned with the *strong requirement* that the Boolean expression for each new task is provided. This is unfortunately not the case for the lifelong RL setting.
>       2. In this paper:
>          1. We give the theoretical analysis for suboptimal value functions in discounted and stochastic tasks. See Sec. 3.2
>          2. We give the theoretical analysis for the case where new tasks are not necessarily expressible as a Boolean expression over past tasks. Additionally, our results are based solely on the quantities the agent learned during each task (learned from the rewards obtained by interacting with the environment). That is, the task decompositions and task specifications are not given to the agent. See Sec. 3.3
>          3. We introduce the SOPGOL algorithm, which leverages logical composition to achieve fast transfer in new tasks. We also give a theoretical analysis of its generalisation ability over task distributions. See Sec. 3.4
>          4. We achieve all this in the lifelong RL setting where the task distribution is *unknown* and possibly *non-stationary*, with the only requirements that the goal space is finite (also required by Tasse et al. 2020) and that the tasks are goal-reaching, i.e defined by desirable goal states (also required by Tasse et al. 2020). Note that before training the agent only receives: the reward bounds (r_MIN, r_MAX), and the dimensionalities of the state and action space to initialise the value function (e.g the DQN), as is standard practice.
>    * These differences could have been clearer and we'll emphasise them in the paper
>
>
> 2. > SOP(., .) is not defined, where the first argument is a set of binary vectors and the second argument is a single binary vector. The result of SOP is B_exp, which is then applied to \tilde{T}_n as if B_exp were a function (and similarly for Q)?
>    * We agree that this should be more prominent and are happy to include a pseudocode.
>    * SOP(\mathcal{T},T) is the classical sum of products method in Boolean logic. Given a list of binary vectors $\mathcal{T}=[T^{'}, T^{''}, …]$, a Boolean expression for a new binary vector T can be obtained by:
>       1. Identifying all rows of T with a 1
>       2. For each such row
>          1. Make a product (conjunction) of all the input variables
>          2. Make the negation of each variable with a 0 in this row
>       3. Take the sum (disjunction) of all these product terms.
>    * The output of SOP is a function $B_{exp}$ that takes in $|\mathcal{T}|$ variables, and applies disjunctions, conjunctions and negations to them according to the Boolean expression obtained above.
>
> 3. > \tilde{T} (the approximation of T, presumably) is not defined within the scope of this theorem -- is it meant to appear in that line or not? I suspect that it was meant to be T, because the theorem does not have any assumptions about the accuracy of \tilde{T} as an approximation of T, and I’m sure that some assumptions would be required to draw any conclusions.
>    * Yes, we use \tilde when we are referring to the approximation of a quantity. We don’t need to make assumptions on how good of an approximation $\tilde{T}$ is, since it directly affects $T_{SOP}$ and $Q_{SOP}$ (which are in turn reflected in the bounds). $T_{SOP}$ and $Q_{SOP}$ are interesting because they represent what the agent thinks (what it inferred with via logical composition), and the optimality of the agent depends on how close they are to T and Q* respectively.
>    * We will make these clearer in the paper.
>
> 4. > In (i), what is the meaning of $\textbf{1}_{T \notin \mathcal{\tilde T_n}}$? From context, this object must be a binary vector. What is it exactly?
>    * It is a number, not a vector. As mentioned, $\textbf{1}$ is the indicator function. So it is equal to 1 if the subscript is true, and 0 if it is false.

---

> > ### Author Response · Authors · 2021-11-21
> > **Reply to Reviewer KyZj [2/4]**
> >
> > 5. > Why are the cardinalities all that we care about? Is there an implication that a task and Q function are only included in those sets if they have been “mastered” with a certain level of accuracy?
> >    * Yes. SOPGOL adds the learned task vector and skill to those sets if it determines that the current task is not expressible in terms of past tasks (See the pseudocode in Appendix B). We will make that clearer in Section 3.4.
> >
> > 6. > How can we draw any conclusions when we have an arbitrary nonstationary distribution of tasks? What prevents a pathological example where we see the same task for the first T steps and then a different one on step T+1, for any T?
> >    * If the same task vector is learned for the first T steps, then that vector and the learned skill will only be added at the first time step.
> >    * Note that SOPGOL adds the learned skill and binary task vector $\tilde T$ to the library only when $\tilde T \neq \mathcal{B}_{exp}(\tilde{\mathcal{T}})$.
> >    * The lower bound is $\lceil log{|\mathcal{G}| \rceil }$ because that is the minimum number of binary vectors that can generate all the other binary vectors (via finite Boolean operations), as can be seen in Table 1 for example. The upper bound is $|\mathcal{G}|$ because that is the dimensionality of the binary vectors.
> >
> >
> > 7. > The experiments are good to illustrate the main ideas, but they are not a substantial standalone contribution, since the domains are very simple and the baseline comparisons are very limited.
> >    * We used the same type of domains (i.e object collection or navigation to goal locations) as previous works (Tasse et al. 2020; Van Niekerk et al., 2019; Abel et al. 2018; Barreto et al. 2018; Saxe et al., 2017; Haarnoja et al., 2018; Hunt et al., 2019). We also used the maxQ baseline (Abel et al. 2018) in our lifelong experiments (Fig 4).
> >    * Just like these previous works, this work assumes an RL method (e.g DQN) that is able to learn goal-reaching tasks in a given environment. Hence the complexity of the environment is not relevant for this work (only the complexity of the goal space), since that only affects how long the regular RL algorithm takes to learn it.
> >    * Consider for example a different algorithm+domain like HAC [1] in the ant domain (see Fig 1 in [1]). Let the goal locations be the same as in our Four-rooms experiment (Sec 4.2). As you can see, fundamentally the high-level tasks are the same (navigating to desired goal locations), but the underlying dynamics are much more complicated. Hence it will take considerably longer for an RL algorithm to learn each task (e.g the y-axis in Fig 4.b will have a much larger order of magnitude), but nothing will change with respect to compositional generalisation. That is Fig 4.a will be exactly the same, since it is solely affected by the task distribution and not dependent on the number of samples required to learn each policy.
> >    * What type of domains or baseline comparisons do you think will add more to the story? We are happy to discuss them.
> >
> >
> > 8. > I was surprised by the new definition of Q-function negation in section 3.2 because the value of the Q-function before negation is not used, except when determining whether the output should “toggle” between Qmax or Qmin. Can you explain this?
> >    * The intuition behind the re-defined negation operator is as follows: since each goal is either desirable or not, the optimal Q(s,g,a) is either Q_MAX(s,g,a) or Q_MIN(s,g,a). Hence, if Q(s,g,a) is closer to Q_MIN(s,g,a), then its negation should be Q_MAX(s,g,a), and vice-versa.
> >
> >
> > 9. > I would like to see a discussion of the relationship between this line of work and propositional AI planning. The composition of goals is very related. It will be easy to find differences, and I think this line of work may offer a different and valuable perspective. I just think that the work would be strengthened by deeply exploring these connections.
> >    * That is an interesting point. We are not doing any planning in this work. Specifically, we are not doing action-level planning (we are using model free RL) nor are we doing task level planning (we are not dealing with sequences of sub-tasks).
> >    * An interesting avenue for future work would be to leverage logical composition of skills in propositional planning to solve tasks defined by sequences of sub-tasks.

---

> > > ### Author Response · Authors · 2021-11-21
> > > **Reply to Reviewer KyZj [3/4]**
> > >
> > > 10. > The premise of the transfer results in this work is that the task specification for a given task is not known, and so we have to compute and use an approximation of it. I am having trouble imagining a scenario where this structured space of possible task specifications is known, but the specification for a given task is not known. Is the idea that for a given task, there may actually be no “true” task specification, and so the approximation is just the “closest” specification that we can find? That would be more compelling, but I don’t think that we can use this interpretation because it would require leaving the class of MDPs defined in this work, and including MDPs that do not necessarily have the good-goal/bad-goal structure with the fixed r0 reward function.
> > >       * Do you mean where the task specification for a given task is known to the agent, or to the environment? If you mean known to the environment, then note that the space of tasks is also structured (they are MDPs). Hence the reward function and transition dynamics are also known. Hopefully you agree that we may not always be able to give the agent the task specification, reward function, or transition dynamics.
> > >       * In this work we are doing model-free RL. That is for each task, the agent in the environment only receives the current state $s$, and the reward $r(s,a)$ for each action $a$ it takes. Therefore, while the agent knows the structured space of possible task MDPs (not just possible task specifications), it does not know the task specification for a given task (nor the reward function and transition dynamics).
> > >       * Side note: Recall that the task specifications here are the binary vectors. Of course, you could imagine future works where for a given task, the agent is also given the task specification as a language statement and it has to figure out what binary vector it corresponds to. You can think of that binary vector as the agents latent representation of the given task.
> > >
> > >
> > > 11. > This comment is about boolean task algebra for RL in general, not just the present work. Though the theoretical contributions here may be interesting in their own right, I am struggling to imagine a practical scenario where it would be preferable to use this framework instead of one where we learn or compute a universal goal-conditioned value function instead.
> > >       * The extended value functions (EVFs) $\bar Q(s,g,a)$ are universal goal-conditioned value functions (UVF). They are called EVFs to indicate that they are defined by the extended reward functions. See a detailed discussion of UVFs in our answer for point 13.
> > >
> > >
> > > 12. > The super-exponential zero-shot generalization advertised in this work and the previous work strikes me as a little bit disingenuous. If I am able to do X, and I’m also able to do Y, then I am automatically also able to do “X or Y”. Is that really generalization? Fundamentally, that is my reading on what’s happening in this work -- we’re learning how to accomplish each of the goals (indirectly, through the tasks, which are sets of goals), and then we’re recombining them in different ways, so that if we know how to achieve goal 1 and goal 2, we also know how to achieve “goal 1 or goal 2.”
> > >       * Logical skill composition is desirable and intuitively simple, but deceptively so:
> > >          1. The tasks: Same environment, many desirable goals.
> > >          2. OR: After learning the skills X and Y, how do we get X OR Y? Is it max{X,Y}, X+Y, etc? Van Niekerk et al., 2019 shows that max{X,Y} is optimal.
> > >          3. AND: After learning the skills X and Y, how do we get X AND Y? Is it min{X,Y}, (X+Y)/2, etc? Haarnoja et al., 2018 shows that (X+Y)/2 is a decent approximation.
> > >       * Hopefully it is now clear why skill composition is non-trivial. Since Tasse et al. shows how to do OR, AND, NOT, and any combination of them optimally, we extend and leverage it in this work for transfer and generalisation in lifelong RL. Please see our answer for point 1 where we emphasize the difference between Tasse et al. and this work.

---

> > > > ### Author Response · Authors · 2021-11-21
> > > > **Reply to Reviewer KyZj [4/4]**
> > > >
> > > > 13. > To elaborate, here is my attempt to describe this line of work through the lens of universal goal-conditioned value function learning:
> > > >     1. > Say that we compute/learn a universal goal-conditioned Q function  $Q(s,g,a)$. Given any goal state $g$, we can do $argmax_a Q(s,g,a)$ to get optimal actions.
> > > >          * What goal-conditioned reward function $r(s,g,a)$ is going to be used to learn the UVF $Q(s,g,a)$? Recall that the environment only gives the agent the current state $s$, and the reward $r(s,a)$ when the agent takes an action $a$.
> > > >          * Usually in the literature UVFs are used for single-goal tasks (e.g pickup a blue key). Here, the environment can give the agent both the current state $s$, the goal-state $g$, and the reward $r(s,g,a)$ when the agent takes an action $a$. This is then used to learn a UVF.
> > > >          * However for many-goal tasks (e.g pickup a key), the environment can’t tell the agent what goal to achieve because there are potentially multiple desirable goals (red keys, blue keys, etc) and the best one to achieve depends on the current state. Hence the agent has to learn how to achieve the best desirable goal from the current state, just from the state-action rewards it receives $r(s,a)$ from the environment. That is why previous works on skill composition (i.e before Tasse et al.) used regular value functions $Q(s,a)$.
> > > >     2. > This line of work says: instead of learning a single goal-conditioned Q function, we could compute/learn multiple goal-conditioned Q functions, one per “task”, where a task is a set of goal states. Each of these Q functions is like an “expert” about some part of the goal space.
> > > >          * Not really, since it is not clear how one would learn such UVFs. Please see the point above.
> > > >          * Tasse et al. says that learning regular value functions is insufficient for zero-shot logical composition, they introduce extended rewards to learn task-based UVFs, and they show that these can be used to achieve zero-shot logical composition.
> > > >     3. > When given a new task $\mathcal{G}$, if we had a universal $Q$, then we could recover an optimal action via $argmax_a \max_{g\in\mathcal{G}} Q(s,g,a)$. This would be “zero-shot” transfer in the sense of this paper. But instead, if we only have the goal-space expert Q functions, we will query the experts for whom  is within their expertise and combine their outputs according to the relationships between  and the expertises. This too will accomplish “zero-shot” transfer. But what is gained over the universal approach?
> > > >          * Not really, since it is not clear how one would learn such UVFs. Please see the point above.
> > > >          * But let’s assume we could learn such a UVF (without extended rewards). Note that in this work and previous works the agent is never given the set of desirable goals $\mathcal{G}_M$ for a given task $M$. Even though we have that information in these simulated toy domains, it is not a practical assumption. This is the same reason why we also have a model of the environment, but we do not give that to the agent (i.e we do model-free RL). The agent has to learn everything from rewards.
> > > >          * Finally, note that Tasse et al. also shows that the same composition procedure holds even when the goal rewards are arbitrary i.e not just good/bad rewards. In this case the tasks cannot even be given by just a set of goals. This means that our work can also be extended to cases where new tasks are preferences over the logical combination of past tasks (e.g by combining it with works on weighed composition like [2,3]).
> > > >
> > > >
> > > > [1] Levy Andrew, et al.; Learning multi-level hierarchies with hindsight.
> > > >
> > > > [2] B. Van Niekerk et al. 2019; Composing value functions in reinforcement learning.
> > > >
> > > > [3] André Barreto, et al.; Fast reinforcement learning with generalized policy updates.

---

> > > > > ### Comment · Reviewer_KyZj · 2021-11-21
> > > > > **Thanks for the thorough response**
> > > > >
> > > > > Thank you very much for the thorough response! It is very helpful.
> > > > >
> > > > > 1. The response to the first question "In brief, ..." is very clear and helpful. I will say more about this below.
> > > > >
> > > > > 2. Thanks for offering SOP pseudocode. I would recommend adding a version of this (ideally, even more clear and formal pseudocode) to the paper appendix. My level of familiarity with Boolean logic is probably at least on par with the average reader, but I had to Google around to understand SOP, and there are surprisingly few clear references available.
> > > > >
> > > > > 3, 4. Thank you for clarifying some of the notation in Theorem 1. So, here is my updated understanding:
> > > > > * $M$ is a new task and $T$ is the true task specification, which is unknown to the agent.
> > > > > * $\bar{Q}^*$ is the optimal extended Q function for T, which is also unknown to the agent.
> > > > > * $\tilde{T}$ is the agent's approximation of $T$.
> > > > > * $\tilde{T}_i$ (and $\tilde{\bar{Q}}_i$) is the agent's approximate task specification (optimal extended value function) for a previously seen task.
> > > > > * $T_{SOP}$ (and $Q_{SOP}$) is the agent's approximate task specification (optimal extended value function) for the new task $M$, in terms of its existing task specifications / value functions. (So, $T_{SOP}$ = $\tilde{T}$?)
> > > > > * $\pi$ is the greedy policy for $Q_{SOP}$ (greedy over both actions and goals).
> > > > > * Conclusion (i) says: if $T_{SOP}$ is correct, and if $T$ is among the previously seen task specifications, then we can obtain a good bound on the value of acting greedily w.r.t. $Q_{SOP}$.
> > > > > * Conclusion (ii) says: even if $T$ is not among the previously seen task specifications, if we additionally assume that the dynamics are deterministic, then we can obtain a similar good bound. (Is there a reason that $Q_{SOP}$ is used in (ii) instead of $Q^\pi$?)
> > > > >
> > > > > Let me know if I've misunderstood anything. Now returning to the relevant part of the response to comment 1:
> > > > > > We give the theoretical analysis for the case where new tasks are not necessarily expressible as a Boolean expression over past tasks. Additionally, our results are based solely on the quantities the agent learned during each task (learned from the rewards obtained by interacting with the environment). That is, the task decompositions and task specifications are not given to the agent. See Sec. 3.3
> > > > >
> > > > > If the new tasks are not expressible as a Boolean expression of past tasks, then the upper bound given by Theorem 1 effectively becomes $\frac{2r_{\Delta}}{1 - \gamma}$. Is this not a trivial bound that would apply to any action-value function?
> > > > >
> > > > > 5, 6. Thank you for clarifying. I think I now understand Theorem 2, but I wonder if it still needs to be said that the distribution over tasks has nonzero support everywhere, or something similar. If there were many possible tasks, but the distribution only ever produced one, then there could be no lower bound, if I understand correctly.
> > > > >
> > > > > 7. I was not suggesting that the experiments need to be extended -- I was just remarking that the main contribution of the paper is theoretical. In other words, I'm not "taking off points" for the current experiments, but I'm also not giving extra points for them. The experiments are a solid but modest validation of the theory.
> > > > >
> > > > > 8. Thanks, this makes sense!
> > > > >
> > > > > 9. I do not think the line between planning and RL is as stark as you suggest. (I know some would disagree with me.) Again, it's easy to find differences, but I encourage exploring connections in future work. The sequence of sub-tasks idea sounds interesting.
> > > > >
> > > > > 10 - 13. Thanks a lot for the responses here. I think this discussion all boils down to a difference in opinion about whether task specifications would be realisitically available to an RL agent in a practical setting. I'll elaborate, but up front I will say that I am not factoring this into my score because I think there are many people (probably a majority) who would agree with your view.
> > > > >
> > > > > My view is that in real applications of RL, whether it's in simulation or the real-world, it would almost always be an engineer who writes down the reward function. So, the reward function would be available in the agent's mind. This is in contrast to the transition function, which could be real world dynamics, and therefore would not necessarily be available in any form to the agent.
> > > > >
> > > > > And I would say the same thing about goals and task specifications. I see how SOPGOL does not _need_ task specifications. But for it to work well, it _does_ need a relatively strong assumption that there exists some good task specifications latent in the reward functions. If the reward functions are specified by a human engineer, it would probably be very easy for that engineer to also specify the task specification. In fact, specifying the task specification would probably be easier than specifying the reward function in many cases.
> > > > >
> > > > > Furthermore, if the engineer supplies a goal (which I also foresee happening), then we could just use universal value functions and avoid tasks altogether.

---

> > > > > > ### Author Response · Authors · 2021-11-21
> > > > > > **Thanks for the quick reply**
> > > > > >
> > > > > > We agree that your suggestions will greatly improve the readability of the paper. We will update it accordingly.
> > > > > >
> > > > > > The bullet points of your understanding of Theorem 1 are correct!
> > > > > >
> > > > > > > Is there a reason that $Q_{SOP}$ is used in (ii) instead of $Q_{\pi}$ ?
> > > > > > * Because we want a tighter bound. (ii) bounds $Q_{SOP}$ directly, while $Q_{\pi}$ bounds it indirectly via $\pi$.
> > > > > >
> > > > > > > If the new tasks are not expressible as a Boolean expression of past tasks, then the upper bound given by Theorem 1 effectively becomes $\frac{2r_{\Delta}}{1-\gamma}$. Is this not a trivial bound that would apply to any action-value function?
> > > > > > * $\frac{2r_{\Delta}}{1-\gamma}$ *only* is indeed a trivial bound, but the bound in (i) include other terms. We indeed get that trivial bound if the new task is not expressible as a Boolean expression of past tasks, which is what you would expect to see. But if the new task is expressible as a Boolean expression of past tasks and the agent correctly infers that expression, then the bound is better, which is what you want to see. That is, the different terms in the bound show how the optimality of the transferred policy changes in different situations. Maybe checking the comparison with the bound from Barreto et al 2018 (Appendix A.4) will make this point clearer.
> > > > > > * However, Theorem 1 (i) is not particularly tight. Theorem 1 (ii) is tighter.
> > > > > >
> > > > > > > I wonder if Theorem 2 still needs to be said that the distribution over tasks has nonzero support everywhere, or something similar. If there were many possible tasks, but the distribution only ever produced one, then there could be no lower bound, if I understand correctly.
> > > > > > * Yes, we implicitly assume that there is a non-zero probability of seeing each task. We will make that explicit.
> > > > > >
> > > > > > > 10 - 13. Thanks a lot for the responses here. I think this discussion all boils down to a difference in opinion about whether task specifications would be realisitically available to an RL agent in a practical setting.
> > > > > > * Not just that. Even if the task specifications are given to the agent, the UVFs approach you suggested still has the fundamental problem of: What goal-conditioned reward function $r(s,g,a)$ is going to be used to learn the UVF $Q(s,g,a)$? Please see again 13.1 and let us know if that makes sense.
> > > > > >
> > > > > > >  My view is that in real applications of RL, whether it's in simulation or the real-world, it would almost always be an engineer who writes down the reward function. So, the reward function would be available in the agent's mind. .... And I would say the same thing about goals and task specifications. I see how SOPGOL does not need task specifications. But for it to work well, it does need a relatively strong assumption that there exists some good task specifications latent in the reward functions.
> > > > > > * Note that there is an entire literature on the difficulty of (and problem with) reward engineering (and mis-specification) [1,2]. Hence that is  not preferable for practical RL. We currently use human-engineered rewards because that is a genuinely hard problem. Ideally, what we want is for a human to judge whether a given trajectory (or end-state for MDPs) satisfies a task or not (yes or no answers) [3]. This could also be in the form of demonstrations, from which the agent needs to figure out the reward function [4]---for example, 0 everywhere and 1 for desired trajectories (i.e latent goal states).
> > > > > > * This highlights that for most tasks we care about---like tasks where a human can judge whether a given trajectory solves or doesn't solve the given task---there exist indeed a good task specification latent in the reward functions.
> > > > > >
> > > > > > > Furthermore, if the engineer supplies a goal (which I also foresee happening), then we could just use universal value functions and avoid tasks altogether.
> > > > > > * Please note that we are considering many-goal tasks. E.g The engineer can say "pickup a key", but that corresponds to many desirable goals in the environment (like the red keys, blue keys, etc). So it is impossible for the engineer to supply a goal.
> > > > > > * If you mean the engineer can supply the set of desirable goals for each task (going back to point 13.3), then beyond our response to point 13.3, that is an extremely strong assumption. To the best of our knowledge, no previous work makes this assumption. That would also invalidate most previous works on transfer learning, since you could make the same argument (learn a UVF then for each task maximize over the given desirable goals). To illustrate with a different line of works, like that of temporal compositions, should one also assume the sequence of sub-goals for each given task is known? That will also invalidate most previous works on temporal compositions.
> > > > > >
> > > > > >
> > > > > > [1] https://openai.com/blog/faulty-reward-functions/
> > > > > >
> > > > > > [2] Dario Amodei et al.; Concrete problems in AI safety
> > > > > >
> > > > > > [3] Paul F Christiano et al.; Deep Reinforcement Learning from Human Preferences
> > > > > >
> > > > > > [4] Andrew Y Ng and Stuart Russell; Algorithms for inverse reinforcement learning

---

> > > > > > > ### Comment · Reviewer_KyZj · 2021-11-22
> > > > > > > **Thank you again!**
> > > > > > >
> > > > > > > Thanks very much for continuing the discussion!
> > > > > > >
> > > > > > > Regarding the bounds in Theorem 1, thanks for confirming my understanding. I see what you're saying about the bounds containing other terms, but those other terms are just indicator functions. It is possible (and easier for me) to understand the bounds in three cases: the task has been previously seen, the task is expressible in terms of previously seen tasks, and the task is neither. The relevant case for claims regarding "few-shot transfer" is the last one; the first two are zero-shot transfer. And in that case, the bound becomes trivial. So revisiting these quotes:
> > > > > > > >If [the task is not expressible], it is able speed up the RL of the current task by using its estimated Q-function (few-shot)
> > > > > > >
> > > > > > > >Tasse et al. 2020 achieved only zero-shot transfer (i.e no few-shot transfer)
> > > > > > >
> > > > > > > >We give the theoretical analysis for the case where new tasks are not necessarily expressible as a Boolean expression over past tasks.
> > > > > > >
> > > > > > > My understanding now is that the few-shot transfer results in this paper (which comprise one of the main deltas on the previous work) are empirical. I see in the analysis and discussion of SOPGOL-transfer that there is some empirical support for SOPGOL performing this kind of few-shot transfer. But the theoretical results for few-shot transfer are trivial, so I do not see them as a contribution over the previous work.
> > > > > > >
> > > > > > > And thanks for continuing an interesting conversation about reward and goal specification!
> > > > > > > >Not just that. Even if the task specifications are given to the agent, the UVFs approach you suggested still has the fundamental problem of: What goal-conditioned reward function is going to be used to learn the UVF?
> > > > > > >
> > > > > > > In my paradigm, the engineer specifies the goal space and the goal-conditioned reward function. (Which is true for all work on goal-conditioned RL that I know about!)
> > > > > > >
> > > > > > > >Note that there is an entire literature on the difficulty of (and problem with) reward engineering (and mis-specification). Hence that is not preferable for practical RL. We currently use human-engineered rewards because that is a genuinely hard problem. Ideally, what we want is for a human to judge whether a given trajectory (or end-state for MDPs) satisfies a task or not (yes or no answers) [3]. This could also be in the form of demonstrations, from which the agent needs to figure out the reward function [4]---for example, 0 everywhere and 1 for desired trajectories (i.e latent goal states).
> > > > > > >
> > > > > > > Yes, this is a very good point. Sometimes reward functions are too difficult to specify. So, I should soften what I said previously. But, if we're learning from a human, why are we restricting the human to giving the robot scalar rewards, and forbidding the human from communicating the intended goal in any form? I just can't imagine this actually happening in an engineering setting. But I could be wrong!
> > > > > > >
> > > > > > > >Please note that we are considering many-goal tasks. E.g The engineer can say "pickup a key", but that corresponds to many desirable goals in the environment (like the red keys, blue keys, etc). So it is impossible for the engineer to supply a goal.
> > > > > > >
> > > > > > > It's not impossible at all:
> > > > > > > ```
> > > > > > > def is_goal_state(state):
> > > > > > >     if any key in state:
> > > > > > >         return True
> > > > > > >     return False
> > > > > > > ```
> > > > > > >
> > > > > > > >If you mean the engineer can supply the set of desirable goals for each task (going back to point 13.3), then beyond our response to point 13.3, that is an extremely strong assumption. To the best of our knowledge, no previous work makes this assumption. That would also invalidate most previous works on transfer learning, since you could make the same argument (learn a UVF then for each task maximize over the given desirable goals).
> > > > > > >
> > > > > > > Okay good, this helps refine what I was trying to say. I don't mean to suggest that transfer learning is unnecessary. Instead I would argue: this line of work on Boolean task algebra relies on the existence of a compact factored representation of the goal space. If none exists, then the approaches will effectively reduce to learning anew for each task. In the work thus far, the factors in the goal space correspond very intuitively to how a human engineer could represent the goals (for example, see the column labels in Table 1). So if it were me, I would just specify the goal space myself in this way. Perhaps there is a case where the factoring found by boolean task algebra is surprising and unintuitive but works well -- I just have trouble imagining this and haven't seen such a case yet. On the point of "no previous work makes this assumption", any work that uses factored goal representations makes this assumption. This is par for the course in the classical planning community, but there's also plenty of older work on factored and relational MDPs with factored goal and reward function representations.

---

> > > > > > > > ### Author Response · Authors · 2021-11-30
> > > > > > > > **Clarifications for theorem 1**
> > > > > > > >
> > > > > > > > Thanks for the continued discussion.
> > > > > > > >
> > > > > > > > > It is possible (and easier for me) to understand the bounds in three cases: the task has been previously seen, the task is expressible in terms of previously seen tasks, and the task is neither. The relevant case for claims regarding "few-shot transfer" is the last one; the first two are zero-shot transfer. And in that case, the bound becomes trivial.
> > > > > > > >
> > > > > > > > > My understanding now is that the few-shot transfer results in this paper (which comprise one of the main deltas on the previous work) are empirical. I see in the analysis and discussion of SOPGOL-transfer that there is some empirical support for SOPGOL performing this kind of few-shot transfer. But the theoretical results for few-shot transfer are trivial, so I do not see them as a contribution over the previous work.
> > > > > > > >
> > > > > > > > * Note that the agent is not told which situation it is in: the task has been previously seen, the task is expressible in terms of previously seen tasks, and the task is neither. It is simply presented with an arbitrary new task after learning n arbitrary tasks, and the bounds show how the optimality of the transferred policy nicely depends on those 3 factors. To make the contributions here clearer, let's compare against the bound of Barreto et al. 2018:
> > > > > > > >     * Our Theorem 1 bounds (we have made (i) stronger as discussed in the comparison below):
> > > > > > > >         * (i) $\|Q^* - Q^{\pi}\|\_\infty \leq \frac{2}{1-\gamma}( (\textbf{1}\_{T \neq T\_{SOP}} + \textbf{1}\_{r \notin {\lbrace{r_g}\rbrace}\_{|\mathcal{G}|}})r\_\Delta + \epsilon)$,
> > > > > > > >         * (ii) If the dynamics are deterministic,
> > > > > > > >   $\|Q^* - Q_{SOP}\|_\infty \leq (\textbf{1} _{T \neq T _{SOP}})r _\Delta + \epsilon,$
> > > > > > > >     * The bound from Barreto et al. 2018 (See Appendix A.4):
> > > > > > > > $ \|Q^*-Q^\pi\|_\infty \leq \frac{2}{1-\gamma}((\textbf{1} _{r \neq r _i} + \textbf{1} _{r _i \notin \lbrace{r _j}\rbrace _n})r _\Delta + \epsilon)$
> > > > > > > >         * The first term of this bound ($\textbf{1}\_{r \neq r\_i}$) requires that reward function of the current task ($r$) be identical to that of a reference task ($r_i$). In Barreto et al. 2018, $r_i$ is taken as the best linear approximation of $r$. In contrast, the first term of Theorem 1.i and ii  ($\textbf{1}\_{T \neq T\_{SOP}}$) only requires the current task to be expressible as a Boolean composition of past tasks.
> > > > > > > >         * The second term of this bound ($\textbf{1}\_{r_i \notin \lbrace{r_j}\rbrace\_n}$) requires that the reference task (the best linear approximation to the current task) is exactly one of the past tasks. In contrast, the second term of Theorem 1.i ($\textbf{1}\_{r \notin \lbrace{r_g}\rbrace\_{|\mathcal{G}|}}$) only requires the current task to have a single desirable goal.
> > > > > > > >         * Finally, Theorem 1.ii is much stronger than this bound and Theorem 1.i, since it says that our transfer learning approach is able to automatically obtain the correct Q-function (i.e with no loss in optimality) when the current task is expressible in terms of past tasks (otherwise, it is suboptimal by only $r_\Delta$, with no dependency on $\frac{2}{1-\gamma}$).
> > > > > > > > * Tasse et al. 2020 is not applicable to this setting (where an arbitrary new task is given after learning n arbitrary tasks), since they are only concerned with how to optimally compose value functions (zero-shot) according to a given Boolean expression.
> > > > > > > > * Finally, to the best of our knowledge, no previous composition work considers the lifelong RL setting where an agent starts with no skill and receives tasks sampled from an unknown distribution (without additional knowledge like base features or true task specifications). In contrast, our approach is able to handle this setting with logarithmic bounds on the number of skills needed to generalise over the task distribution (Theorem 2).
> > > > > > > >
> > > > > > > > Thanks for the discussion about rewards and goal specification. We agree that your suggestion makes sense in the planning community, but as you mentioned earlier, this is not the view of most of the RL community (or at the very least of the transfer learning and lifelong learning sub-community).

---

### Official Review · Reviewer_AVki · 2021-10-30

**Correctness:** 2
**Technical Novelty And Significance:** 4
**Empirical Novelty And Significance:** 3
**Recommendation:** 5
**Confidence:** 2

**Main Review:**

Goal-based tasks: I think this is an interesting subset of RL problems, and that it is great to study this class of problems.  However, not mentioning this limitation in the abstract (and not even in intro) seems very odd; ideally “goal-based” would be not only in the abstract and intro, but also in the title, since there is such a fundamental difference between looking at RL in general and at this subset of RL.  It was odd to read the title, abstract, and intro, and then have my expectations of what the paper was about completely disrupted in the third sentence of Section 2.

Section 2 issues:

- r_0 is neither formally defined nor intuitively explained (ideally it should have both a definition and an explanation), so its significance and the difference between it and r is unclear.

- ”By penalising the agent for achieving…” are there typos in this sentence?  I cannot parse it.  Is \mathcal G— a new symbol (with the additional “—”)?  What does “r_min&” mean in this context?

- Definitions 1-2, extended reward function and extended Q-value function: I’m a little confused by what these are supposed to do.  Are we changing the problem/MDP, replacing the old reward function with the extended one?  Or are we changing the agent to utilize these concepts, while the problem remains the same?  Ideally, the text before/after these definitions would better explain the purpose of the definitions, avoiding this potential reader confusion.

- “the extended reward function has the effect of driving the agent to learn how to
separately achieve all desirable goals.” This is confusing to me; it seems like this may have the effect of driving the agent to learn how to achieve only some of the goals (perhaps only one), not all the goals.  (I suspect the confusion is due to one of the issues above, and will be cleared up by fixing and clarifying one of those issues.)

- Definition 3: since r_0 and r_[some task] have not been defined, I cannot parse definition 3.

- Because of the cumulative effect of the Section 2 issues above (particularly the notation issues), I am not able to give constructive feedback on the theoretical contributions of Section 3.  I found Section 3 difficult to follow, but could not determine if the problem was Section 3, or some of my confusions with Section 2.  While Section 2 is only the background section, it establishes some notation and concepts that likely are not standard for many readers with an RL background.  Since Section 3 proceeds to build upon Section 2, I think that the issues with Section 2 will significantly harm the readability of Section 3 for many readers.

[End Section 2 issues]

4.1 experiments: The experiments used only 4 runs, which is concerning for any experimental claims.  The claim is that SOPGOL is exhibiting transfer learning (4.1), and the evidence is that SOPGOL starts with much stronger performance than the baseline.  However, 4 runs is simply not a sufficient amount of data to make this claim.  I would encourage the authors to take the time to run 30+ runs in the future to make this claim more substantial.  Statistical significance testing would also be nice, but, if something like a t-test is used, much more data is needed anyway, since the t-test normality assumption is not reasonable with 4 runs (but might be reasonable with 30+, because of the CLT).

Minor Figure 4 formatting issue: I was thrown off by the lack of spacing between the two subfigures’ captions, and for a couple minutes was trying to understand it as a single paragraph starting as: “(a) Number of policies required to learn and store (b) Number of samples required…”.  The correct way of reading it, as two separate paragraphs side-by-side, is obvious once one sees it, but the way the paragraphs are set so close together makes me think that other readers may get confused as well.  (Also, the first lines of the two paragraphs are unfortunate in that they almost make sense when read in the wrong way; the wrong way of reading it only turns into obvious nonsense in the second or third lines.)  I suggest adding a bit of horizontal spacing between the sub-captions to help avoid this issue (if negative horizontal space was used here, please do not do that).

Minor notation edits:

- inconsistency in MDP tuple ordering (sometimes discount parameter is last, other times reward function is last).

- In theorem 1, T_SOP and \bar Q_SOP definitions use the equals symbol, should they use \coloneqq instead?

**Update after rebuttal:**

> We think of transfer along these lines of works [1,2,3]. For example, if an agent can learn to “pickup yellow objects”, then we would like it to transfer this knowledge to “pickup yellow keys” or even “pickup keys that are not yellow”. We are curious as to the kinds of tasks that you believe we should cover.

I’m not suggesting changing the tasks at all, but merely suggesting that the fact that the paper exclusively studies goal-based tasks be mentioned earlier (which you already said you would do, so no need to discuss this further).

Or am I misunderstanding?  Is the reasoning that most or all of transfer learning in RL can be framed as goal based tasks?  If so, it’s not so much a limitation that necessarily needs to be mentioned earlier; instead, in that case, I think the fix is to explain that most/all of transfer learning can be framed as goal-based tasks in the first paragraph of Section 2.

> Definitions 1-2...We will make that clearer in the paper

Great, thank you.

Point 3 of rebuttal: The revised version is much more clear, thank you.

More Definition 3 feedback: I still struggled with Definition 3 despite the clarifications above.  Was a “task” ever formally defined?  (See my clarity concerns below, the setting is still not clear to me.)  I was also confused about how a task could be “bounded”.

Two more points regarding lack of clarity in the setting:

1) > During each episode, the agent samples a goal to reach from its goal buffer. If it successfully reaches that goal, then the extended reward it receives is just the regular task reward from the environment. If it reaches a different terminal goal state, then the extended reward it receives is the large penalty. Hence the agent only gets the high task rewards when it reaches the goal it is trying to reach. This drives it to learn to achieve all the goals it encounters. Please see Fig 4 of Tasse et al. if that helps.

It’s not clear from the paper that each episode has the agent sampling a new goal for the task.  It becomes slightly more clear in Section 3 (but still isn’t completely clear then), but I think a big source of ambiguity and confusion in Section 2 is that this is not explained well.  More generally, the explanation of the relationships between goals, tasks, episodes, and the distribution of tasks needs work to improve clarity.

2)  3.1, task space size being 2^|G|: it’s not clear to me exactly where this is coming from.  Is it because each goal can be either r_min or r_max for a given task?  More clarity in section 2 would probably help avoid this confusion.

Summary of update:  Some concerns have been addressed (particularly the promised change in phrasing weakening the claims regarding the empirical results in 4.1), and I am increasing my score slightly, but clarity is still a major concern for me.  Because I am not understanding the setting and theory well, my confidence in my assessment is low, and I recommend disregarding my assessment if the other reviewers' clarity concerns are addressed.


**Summary Of The Paper:**

The authors study goal-based lifelong RL.  They leverage logical composition to create an algorithm for this setting, with the goal of better generalization.  They provide theoretical bounds for their approach, and provide empirical evidence that their approach generalizes and transfers well in practice.

**Summary Of The Review:**

This paper is fascinating and well-motivated.  However, a few notation and clarity issues made it difficult to follow Sections 2-3, and I am also concerned about the statistical significance of the authors’ claims in 4.1.

---

> ### Author Response · Authors · 2021-11-22
> **Reply to Reviewer AVki [1/3]**
>
> Thank you for your careful review of our paper. We hope that the following points address your concerns.
>
>
> 1. Goal-based tasks: I think this is an interesting subset of RL problems, and that it is great to study this class of problems. However, not mentioning this limitation in the abstract (and not even in intro) seems very odd; ideally “goal-based” would be not only in the abstract and intro, but also in the title, since there is such a fundamental difference between looking at RL in general and at this subset of RL. It was odd to read the title, abstract, and intro, and then have my expectations of what the paper was about completely disrupted in the third sentence of Section 2.
>    * Thanks for picking up on that. We will specify in the abstract and introduction that we are considering goal-based tasks.
>    * We think of transfer along these lines of works [1,2,3]. For example, if an agent can learn to  “pickup yellow objects”, then we would like it to transfer this knowledge to “pickup yellow keys” or even  “pickup keys that are not yellow”. We are curious as to the kinds of tasks that you believe we should cover.
>
>
> 2. r_0 is neither formally defined nor intuitively explained (ideally it should have both a definition and an explanation), so its significance and the difference between it and r is unclear.
>    * As mentioned in Section 2, $r_0$ is the reward function of the background MDP (which represents the environment the agent is in). As shown by Equation (1), $r$ is the reward function of a specific task MDP, and it is defined by the environment rewards ($r_0$) everywhere except at goal states. Our approach is agnostic to the choice of $r_0$. We only care that the task rewards $r$ have that structure. That is, the tasks have the same rewards at non-goal states (e.g 0 everywhere and 1 at desirable goal states). This structure represents the setting where an agent is in an environment and receives various many-goal tasks throughout its lifetime.
>
>
> 3. ”By penalising the agent for achieving…” are there typos in this sentence? I cannot parse it. Is \mathcal G— a new symbol (with the additional “—”)? What does “r_min&” mean in this context?
>    * “—” is a punctuation mark similar to brackets.  We can change to brackets if you believe this would be clearer. We use it to help the reader understand which part of the extended reward function is “penalising the agent” ($\bar r_{MIN} \text{ if } g \neq s \in \mathcal{G}$). Thanks for picking up on the "&", that was a typo.
>
>
> 4. Definitions 1-2, extended reward function and extended Q-value function: I’m a little confused by what these are supposed to do. Are we changing the problem/MDP, replacing the old reward function with the extended one? Or are we changing the agent to utilize these concepts, while the problem remains the same? Ideally, the text before/after these definitions would better explain the purpose of the definitions, avoiding this potential reader confusion.
>    * They are changing the agent to utilize these concepts. Essentially, Tasse et al. 2020 introduces extended value functions (EVFs) as the knowledge agents need to learn in each task in order for logical composition to work. They also show that when an agent learns an EVF, they can still recover the desired task policy from it similarly to regular value functions by maximising over goals and actions: $\pi(s) \in argmax_a \max_g  \bar Q (s, g, a)$
>    * We will make that clearer in the paper
>
>
> 5. “the extended reward function has the effect of driving the agent to learn how to separately achieve all desirable goals.” This is confusing to me; it seems like this may have the effect of driving the agent to learn how to achieve only some of the goals (perhaps only one), not all the goals. (I suspect the confusion is due to one of the issues above, and will be cleared up by fixing and clarifying one of those issues.)
>    * We hope that the responses above helped to clarify this point.
>    * During each episode, the agent samples a goal to reach from its goal buffer. If it successfully reaches that goal, then the extended reward it receives is just the regular task reward from the environment. If it reaches a different terminal goal state, then the extended reward it receives is the large penalty. Hence the agent only gets the high task rewards when it reaches the goal it is trying to reach. This drives it to learn to achieve all the goals it encounters. Please see Fig 4 of Tasse et al. if that helps.
>    * The main point here is that learning extended value functions enables one to achieve logical composition provably.
>
>
> 6. Definition 3: since r_0 and r_[some task] have not been defined, I cannot parse definition 3.
>    * Please see our response to point 2 about r_0 and r_[some task]. Note that $r_M$ is the reward function for a task $M \in \mathcal{M}$.

---

> > ### Author Response · Authors · 2021-11-22
> > **Reply to Reviewer AVki [2/3]**
> >
> > 7. Because of the cumulative effect of the Section 2 issues above (particularly the notation issues), I am not able to give constructive feedback on the theoretical contributions of Section 3. I found Section 3 difficult to follow, but could not determine if the problem was Section 3, or some of my confusions with Section 2. While Section 2 is only the background section, it establishes some notation and concepts that likely are not standard for many readers with an RL background. Since Section 3 proceeds to build upon Section 2, I think that the issues with Section 2 will significantly harm the readability of Section 3 for many readers.
> >    * We hope that our responses above have clarified your issues here. Please note that we use the same notations as previous works (e.g [1,2,3]), but we are happy to clarify any other notation that was not sufficiently clear. Please see point 10 for a summary of Section 3
> >
> >
> > 8. 4.1 experiments: The experiments used only 4 runs, which is concerning for any experimental claims. The claim is that SOPGOL is exhibiting transfer learning (4.1), and the evidence is that SOPGOL starts with much stronger performance than the baseline. However, 4 runs is simply not a sufficient amount of data to make this claim. I would encourage the authors to take the time to run 30+ runs in the future to make this claim more substantial. Statistical significance testing would also be nice, but, if something like a t-test is used, much more data is needed anyway, since the t-test normality assumption is not reasonable with 4 runs (but might be reasonable with 30+, because of the CLT).
> >    * We agree with your points, but unfortunately these experiments are very expensive to run. Training a single DQN agent (for our approach or the baseline) takes about a million samples (that is several hours of training).
> >    * Just like previous theoretical works [1,2,3], these experiments are mainly to demonstrate the theoretical results and to have some useful discussions around them.
> >    * Our main claims/contributions are the theoretical results. We hope you agree that they are significant (please also see point 10).
> >    * Finally, please note that the lifelong RL experiments (Fig. 4) are averaged over 25 runs. This is only possible because we are using a gridworld domain here, and still the time complexity is crazy (4 graphs per figure x 50 tasks x 25 runs x Average timesteps to learn each task).
> >
> >
> > 9. I suggest adding a bit of horizontal spacing between the sub-captions to help avoid this issue (if negative horizontal space was used here, please do not do that).
> >    * Thank you for the suggestion. We will update the paper accordingly.

---

> > > ### Author Response · Authors · 2021-11-22
> > > **Reply to Reviewer AVki [3/3]**
> > >
> > > 10. A few notation and clarity issues made it difficult to follow Sections 2-3, and I am also concerned about the statistical significance of the authors’ claims in 4.1.
> > >       * We hope the previous points clarified the notational issues. For better clarity, we would like to summarize and emphasise the importance of this work and our contributions.
> > >       * At a high level, the motivation for this work is if we are ever going to have lifelong robots acting in the real world to solve tasks we care about, then they need at the very least to be able to solve many-goal tasks in a sample efficient way. Consider for example a domestic robot that needs to “cook various types of food”, “buy groceries”, “buy groceries or deliver water and food to the construction site”, etc. Hence we care about agents in an environment that receive various many-goal tasks throughout their lifetime.
> > >       * In brief, our approach says the following: When an agent is presented with a new task, it is able to learn what task it is required to solve in a supervised way (by learning its binary specification from rewards). It is then able to generate a boolean expression to determine whether or not that task is solvable zero-shot using its previously acquired skills (learned Q-functions). If yes, then it is able to produce the Q-function that solves the current task without any further learning. If not, it is able speed up the RL of the current new task by using its estimated Q-function (few-shot).
> > >       * To the best of our knowledge, this is the first work that achieves such a degree of transfer learning and generalization. For example,
> > >          1. Tasse et al. 2020 achieved only zero-shot transfer (i.e no few-shot transfer) when the *optimal* value functions are learned with the *strong requirement* that the Boolean expression for each new task is provided. This is unfortunately not the case for the lifelong RL setting.
> > >          2. Barreto et al. 2020, achieved only few-shot transfer for non-base tasks with the *strong requirement* that the reward function of each task M is well approximated by a linear function (i.e r_M(s,a) ~= theta(s,a)*w_M). This is unfortunately not the case for many-goal tasks sampled from an unknown distribution in the lifelong RL setting. E.g There is no linear combination of  [0,0,1,1] and [0,1,0,1] that gives a good approximation for [0,1,1,0] (i.e their exclusive OR).
> > >       * In this paper:
> > >          1. We give the theoretical analysis for logical composition with suboptimal value functions in discounted and stochastic tasks. See Sec. 3.2
> > >          2. We give the theoretical analysis for the case where new tasks are not necessarily expressible as a Boolean expression over past tasks. Additionally, our results are based solely on the value functions and binary vectors the agent learned during each task (learned from the rewards obtained by interacting with the environment). That is, the task decompositions and task specifications are not given to the agent. See Sec. 3.3
> > >          3. We introduce the SOPGOL algorithm, which leverages logical composition to achieve fast transfer in new tasks. We also give a theoretical analysis of its generalisation ability over task distributions. See Sec. 3.4
> > >          4. We achieve all this in the lifelong RL setting where the task distribution is *unknown* and possibly *non-stationary*, with the only requirements that the goal space is finite (also required by Tasse et al. 2020) and that the tasks are goal-reaching, i.e defined by desirable goal states (also required by Tasse et al. 2020). Note that before training the agent only receives: the reward bounds (r_MIN, r_MAX), and the dimensionalities of the state and action space to initialise the value function (e.g the DQN), as is standard practice.
> > >
> > > [1] Tasse et al. 2020; A Boolean task algebra for reinforcement learning.
> > >
> > > [2] Barreto et al. 2018; Transfer in deep reinforcement learning using successor features and generalised policy improvement.
> > >
> > > [3] Abel et al. 2018; Policy and value transfer in lifelong reinforcement learning.

---

> > > > ### Comment · Reviewer_AVki · 2021-11-24
> > > > **Quick Question**
> > > >
> > > > I will get to the rest of these points soon, and possibly reevaluate my score, but wanted to ask quick question before the end of the phase where authors can respond:
> > > >
> > > > >We agree with your points, but unfortunately these experiments are very expensive to run. Training a single DQN agent (for our approach or the baseline) takes about a million samples (that is several hours of training).
> > > >     Just like previous theoretical works [1,2,3], these experiments are mainly to demonstrate the theoretical results and to have some useful discussions around them.
> > > >     Our main claims/contributions are the theoretical results. We hope you agree that they are significant (please also see point 10).
> > > >     Finally, please note that the lifelong RL experiments (Fig. 4) are averaged over 25 runs. This is only possible because we are using a gridworld domain here, and still the time complexity is crazy (4 graphs per figure x 50 tasks x 25 runs x Average timesteps to learn each task).
> > > >
> > > > This all makes sense.  However, I'm still concerned that the paper makes claims that are not supported by the data.  If I increased my score (no promises, I want to spend more time on the rebuttal), would you be willing to tone down the claims?  Specifically, you could say something like "Initial evidence (limited by 4 runs) suggests that transfer learning is occurring", or something like that, and then emphasize that it's primarily a theoretical paper with a small empirical contribution (or at least state that 4.1 is a small "initial evidence"-type of contribution).  If you are able to respond before the end of the phase, that would be appreciated.  Thank you.

---

> > > > > ### Author Response · Authors · 2021-11-24
> > > > > **Yes**
> > > > >
> > > > > That is a fair point that will help make the contributions clearer. We will update the paper as suggested, making it clear that the function experiments are initial evidence to demonstrate the theoretical results. Thanks!

---

> > > > > > ### Comment · Reviewer_AVki · 2021-11-25
> > > > > > **Thank you**
> > > > > >
> > > > > > Thank you for your hard work.  I have updated my review; I have increased my score slightly but still have significant concerns about clarity.  However, as indicated in that update, my confidence in my assessment is low.

---

### Official Review · Reviewer_1zPe · 2021-11-01

**Correctness:** 3
**Technical Novelty And Significance:** 2
**Empirical Novelty And Significance:** 2
**Recommendation:** 5
**Confidence:** 3

**Main Review:**

### Strengths
* The paper is generally well written and easy to follow, and explanations are intuitive.
* The chosen research problem, lifelong and multi-task RL, is both important and insufficiently studied.
* I really like the idea of having a task-composition logic. I think that this is an immensely interesting research direction.
* I like the theoretical results, even though they are a little unsurprising.

### Weaknesses
* Some important details are missing from the main paper: the exact algorithm proposed, and the exact setup of the experiments.
* The paper's experimental section is fairly weak in that all experiments are on the same, fairly simple task.
* It is not clear if the experiments are "fair", i.e., if the algorithms play on an equal playing field.

### Detailed questions/comments
* While the paper is properly anonymized in theory, I looked up "Nangue Tasse et al (2020)". The fact that they use near-duplicate language strongly suggests the same authors (or a violation of attribution). Please re-write to avoid this suspicion.
* Some notations are off or ambiguous (like in (2) - what does "s \neq s \in G" mean? Similarly in the theorems - why are there two inequalities combined with an equal sign?)
* It is completely unclear to me how the algorithm interacts with the chosen DQN model in detail. In particular, this sentence baffles me and I suspect that a lot is going on behind the scenes: "Having learned the optimal extended value functions for our base tasks, we can now leverage logical composition to do transfer learning on test tasks."
* Is the task composition provided to the learners in some way, or do they have to trial-and-error the task composition?
* From the look of the plots, a pure DQN may get competetive using sufficient samples? If so, please train with more data/longer. This would spin the paper more towards a data efficiency argument.

**Summary Of The Paper:**

The paper introduces an approach of logical decomposition of tasks as a particular set of MDPs. This logical decomposition allows to define a boolean logic over tasks and q-functions. Using this boolean logic, the paper continues in proving two boundaries for generalizing to new, unencountered tasks. The first states a boundary on performance on a new task when following a composition of previously learned policies. The second states that, to solve problems composed of k atomic tasks optimally, at most k skills will ever be required. The paper continues define a lifelong learning algorithm that relies on this logic to form a set of base skills. This algorithm is then evaluated on a multi-task and lifelong learning task, where it decidedly outperforms a standard deep q-learner.

**Summary Of The Review:**

The targeted problem - a lack of multi-task and lifelong learning capabilities of most RL algorithms - is an extremely important one, as it resembles real-life applications a lot more closely. Additionally, the idea of finding a decomposition of multiple tasks via boolean logic is highly appealing, as it allows structural abstraction in RL, another important but underexplored question. Unfortunately, the paper's contributions are a little underwhelming. The logical decomposition is nice but fairly straightforward. The theorems provide an interesting, but also not groundbreaking insight. And finally, the experiments are unclear in execution, only on one domain, and therefore unconvincing. Overall I really like the idea, but the paper needs substantial work to become a full contribution.

---

> ### Author Response · Authors · 2021-11-14
> **Reply to Reviewer 1zPe [1/2]**
>
> Thank you for your careful review of our paper. We hope that the following points address your concerns.
>
>
> 1. Some important details are missing from the main paper: the exact algorithm proposed, and the exact setup of the experiments. It is completely unclear to me how the algorithm interacts with the chosen DQN model in detail.
>    * We included a description of SOPGOL in Section 3.4, but unfortunately could only include the pseudocode in Appendix B for space reasons. We also included additional function approximation details in Appendix C. Which specific details do you feel are missing? We are happy to clarify and include them in the text.
>    * Here is a summary:
>       1. The agent receives the reward bounds $(r_{MIN}, r_{MAX})$, and the dimensions of the state space |S| = 56x56x3, the goal space |G|=15, and the action space |A| of the environment. These are used to initialise |G| DQNs to learn the extended Q-functions, $\bar Q(s,g,a)$.
>       2. The main parts of SOPGOL are:
>          1. The extended Q-function $\bar Q$, task vector T, and goal buffer G are initialised.
>          2. At the beginning of each episode, the agent computes $T_{SOP}$ and $\bar Q_{SOP}$ for the learned task vector T using the SOP method and its library of task vectors and extended Q-functions.
>          3. If $T_{SOP} = T$, the agent acts using e-greedy with  $\bar Q_{SOP}$. Otherwise, the agent acts using e-greedy with $max${ $\bar Q_{SOP}, \bar Q$ }.
>          4. If $T_{SOP} \neq T$, the agent updates $\bar Q$ for each goal in the goal buffer using Deep-Q learning.
>          5. If the agent reaches a terminal state s, it adds it to the goal buffer and updates T (in a supervised manner as per Equation 6).
>          6. Learning stops after n episodes or when the agent has reached a desired level of optimality.
>
>
> 2. The paper's experimental section is fairly weak in that all experiments are on the same, fairly simple task.
>    * Please note that this work assumes an RL method (e.g DQN) that is able to learn goal-reaching tasks in a given environment. Hence the complexity of the environment is not relevant for this work (only the complexity of the goal space), since that only affects how long the regular RL algorithm takes to learn it.
>    * We chose the PickUpObj domain because it illustrates what we care about in this work. It is an environment with many-goal tasks which can be learned with an off-the-shelf RL algorithm like deep-Q learning (but in a very sample inefficient way and with poor generalisation over a task distribution).
>    * Which other environment do you prefer that has many-goal tasks which are solvable with an off-the-shelf RL algorithm? We are happy to update our experiments with it.
>
>
> 3. It is not clear if the experiments are "fair", i.e., if the algorithms play on an equal playing field.
>    * In all our experiments, we used the same RL algorithm in SOPGOL and the baseline. Both were trained on the same tasks with the same hyperparameters for the RL algorithm. We will emphasise this in the paper.
>    * Which additional experiment details do you feel are missing? We are happy to clarify and include them.
>
>
> 4. While the paper is properly anonymized in theory, I looked up "Nangue Tasse et al (2020)". The fact that they use near-duplicate language strongly suggests the same authors (or a violation of attribution). Please rewrite to avoid this suspicion.
>    * We used similar language and notations from previous works [1,2,3] where relevant. We emphasize that:
>       1. Our approach of defining tasks as sharing the same background MDP is different from [1] (in addition to considering stochastic discounted MDPs). We believe this is a more natural definition of tasks because it highlights that we are ultimately interested in agents that exist in an environment and that receive various many-goal tasks throughout their lifetime.
>       2. For clarity, we summarise the main results of [1] in Proposition 1 in the same form as our Proposition 2.
>       3. We define lifelong RL similarly to [2], but extend it to include non-stationary task distributions (see Section 3.1).
>       4. We use similar language and notations as [3] for our theorems and theoretical results, and simplify our bounds further (using indicator functions instead of norms) to make them clearer. This helps to further consistent theory notations across the literature and makes it easy to compare our bounds with that of [3] (see Appendix A.4).
>    * Which specific parts do you feel were not correctly attributed? We believe we emphasised which work we were building on in Sections 2.1 and 3.2. We are happy to clarify and correct them.
>
>
> [1] G. Nangue Tasse et al., A Boolean task algebra for reinforcement learning.
>
> [2] D. Abel et al., Policy and value transfer in lifelong reinforcement learning.
>
> [3] Andre Barreto et al., Transfer in deep reinforcement learning using successor features and generalised policy improvement.

---

> > ### Author Response · Authors · 2021-11-14
> > **Reply to Reviewer 1zPe [2/2]**
> >
> > 5. Some notations are off or ambiguous (like in (2) - what does "s \neq s \in G" mean? Similarly in the theorems - why are there two inequalities combined with an equal sign?)
> >    * These are mathematical statements.
> >       1. "g \neq s \in G" means  “g \neq s and s \in G"
> >       2. The notation in Theorem 2 is in the standard literature (e.g see Appendix A of [3]) and just means each inequality and equality is true. We are happy to split the equality from the inequalities for more clarity.
> >
> >
> > 6. Is the task composition provided to the learners in some way, or do they have to trial-and-error the task composition?
> >    * The agent infers it by learning the task binary vector and using the SOP method. In summary:
> >    * The task binary vector T is learned in a supervised manner using terminal rewards. That is, when the agent reaches a terminal state s,
> >         $T(s) ← 1_{(r=r_{MAX})}$
> > where r is the reward the agent received in s (Equation 6).
> >    * SOP is the classical sum of products method in Boolean logic (we are happy to include a pseudo-code version if that would help). Given a list of binary vectors $\mathcal{T}=[T^{'}, T^{''}, …]$, a Boolean expression for a new binary vector T can be obtained by:
> >       1. Identifying all rows of T with a 1
> >       2. For each such row
> >          1. Make a product (conjunction) of all the input variables
> >          2. Make the negation of each variable with a 0 in this row
> >       3. Take the sum (disjunction) of all these product terms.
> >
> >
> > 7. From the look of the plots, a pure DQN may get competitive using sufficient samples? If so, please train with more data/longer. This would spin the paper more towards a data efficiency argument.
> >    * As mentioned in point 2, this work assumes an RL method (e.g DQN) that is able to learn goal-reaching tasks in a given environment. Hence we expect DQN to eventually learn each task, as can be seen from the plots. The aim of this work is to leverage composition to improve on the sample-efficiency of learning by orders of magnitude (e.g Fig 3.a), and most importantly to achieve generalisation over a task distribution (see Fig 4).

---

> > > ### Comment · Reviewer_1zPe · 2021-11-18
> > > **Response**
> > >
> > > Re:
> > > 1. I have a problem with a paper if something as essential as how the algorithm actually trains a model is moved to the appendix. The appendix should only act to add non-essential information - otherwise it would just be a way to get around the page limit. Unfortunately, I believe that the way that the algorithm interacts with the DQN model is essential.
> > > 2. "Please note that this work assumes an RL method (e.g DQN) that is able to learn goal-reaching tasks in a given environment. Hence the complexity of the environment is not relevant for this work (only the complexity of the goal space), since that only affects how long the regular RL algorithm takes to learn it." - I am not sure I believe this. Training complexities for a particular task/model combination oftentimes translate into challenges for an entire ML system. Another experiment on a completely different problem - maybe even one where there is no clear logical decomposition - would have gone a long way to make the paper more convincing.
> > > 3. It is not clear if the experiments are "fair", i.e., if the algorithms play on an equal playing field -> this is where having more details in the paper is helpful, because it would alleviate these questions immediately.
> > > 5. "The notation in Theorem 2 is in the standard literature (e.g see Appendix A of [3]) and just means each inequality and equality is true. We are happy to split the equality from the inequalities for more clarity." -> It is still bad notation. The adequate symbol would probably be a "∧" for logical conjunction.
> > > 6. That is very helpful. Please include a version of this in the main paper.
> > >
> > > Please address some of these issues, and I will increase my score.

---

> > > > ### Author Response · Authors · 2021-11-19
> > > > **Response to Reviewer 1zPe**
> > > >
> > > > We agree that these will greatly improve the readability of the paper. For points 1, 3, and 5, we will include the relevant parts of our responses in the paper. We are happy to make the change mentioned in point 4. For point 2:
> > > > * As we showed in the paper, our approach works for any goal-reaching problem with finite goal space, which is a very large class of problems of interest and the same one considered by previous works (Tasse et al. 2020; Van Niekerk et al., 2019; Abel et al. 2018; Barreto et al. 2018; Saxe et al., 2017; Haarnoja et al., 2018; Hunt et al., 2019). These works also use the same type of domains for their experiments (i.e object collection or navigation to goal locations).
> > > > * Consider for example a different algorithm+domain like HAC [1] in the ant domain (see Fig 1 in [1]). Let the goal locations be the same as in our Four-rooms experiment (Sec 4.2). As you can see, fundamentally the high-level tasks are the same (navigating to desired goal locations), but the underlying dynamics are much more complicated. Hence it will take considerably longer for an RL algorithm to learn each task (e.g the y-axis in Fig 4.b will have a much larger order of magnitude), but nothing will change with respect to compositional generalisation. That is Fig 4.a will be exactly the same, since it is solely affected by the task distribution and not dependent on the number of samples required to learn each policy. Please let us know if we have missed the gist of your question.
> > > >
> > > >
> > > > Finally, we would like to address the following point you made earlier:
> > > > > Unfortunately, the paper's contributions are a little underwhelming. The logical decomposition is nice but fairly straightforward. The theorems provide an interesting, but also not groundbreaking insight.
> > > >
> > > >
> > > > * While the logical decomposition is straightforward (which is desirable), it is meaningless without the ability to successfully compose skills according to it. Theorem 1 gives guarantees on the usefulness of the composed skills and Theorem 2 gives guarantees on the generalization ability of the agent. We are glad that they make intuitive sense (which is desirable), but they are by no means trivial. To the best of our knowledge, this is the first work that achieves such a degree of transfer learning and generalization.
> > > >    * Previous work, Tasse et al. 2020, achieved only zero-shot transfer (i.e no few-shot transfer) when the *optimal* value functions are learned with the *strong requirement* that the Boolean expression for each new task is provided. This is unfortunately not the case for the lifelong RL setting.
> > > >    * Previous work, Barreto et al. 2020, achieved only few-shot transfer for non-base tasks with the *strong requirement* that the reward function of each task M is well approximated by a linear function (i.e r_M(s,a) ~= theta(s,a)*w_M). This is unfortunately not the case for many-goal tasks sampled from an unknown distribution. E.g There is no linear combination of  [0,0,1,1] and [0,1,0,1] that gives a good approximation for [0,1,1,0] (i.e their exclusive OR).
> > > >    * In contrast, we achieve both zero-shot and few-shot transfer with the only requirements that the goal space is finite (also required by previous works) and that the tasks are goal-reaching, i.e defined by desirable goal states (also required by previous works). Note that before training the agent only receives: the reward bounds (r_MIN, r_MAX), and the dimensionalities of the state and action space to initialise the DQN, as is standard practice.
> > > >
> > > >
> > > > [1] Levy Andrew, et al. Learning multi-level hierarchies with hindsight.

---

> > > > > ### Comment · Reviewer_1zPe · 2021-11-22
> > > > > **Looking forward to next version of the paper**
> > > > >
> > > > > I am not (yet) convinced of the claims the authors make, but I will re-evaluate after having read a new version of the paper.

---

> > > > > > ### Author Response · Authors · 2021-11-26
> > > > > > **Paper revised**
> > > > > >
> > > > > > We have updated the paper to reflect your suggestions. As we are nearing the end of the discussion period, do you have any outstanding concerns in light of our response? We are happy to address them.

---

> > > > > > > ### Comment · Reviewer_1zPe · 2021-11-29
> > > > > > > **Happy with the modifications**
> > > > > > >
> > > > > > > While some of my more general concerns still exist (about the relevance of the empirical section, inparticular), I am convinced by some of the other modifications. I have increased my score.

---

### Official Review · Reviewer_ovqB · 2021-11-02

**Correctness:** 3
**Technical Novelty And Significance:** 3
**Empirical Novelty And Significance:** 3
**Recommendation:** 8
**Confidence:** 3

**Main Review:**

Positives:

- The idea of using logical composition to autonomously determine whether a given task can be immediately solved using existing policies or whether a new policy has to be learned and added to the policy library is interesting.

- Overall, the paper is well-written and it is easy to follow (if typos are not taken into account).

- It is nice to see that the paper contains theoretical results that provide bounds (i) on the performance of transferred policy on a new task and (ii) on the necessary and sufficient number of tasks that need to be learned by the agent so that it can generalize over a distribution. The proofs of the theorems also seem sound.

- The experiments section of the paper provides comprehensive results, including results on transfer after pretraining on a set of tasks (both base and arbitrary) and on lifelong transfer scenarios where the agent builds the skill set from scratch, verifying the theoretical results. Using three different task distributions in the lifelong RL setting is also a nice way to show how the algorithm would perform in best-case, worst-case and average-case scenarios.


Concerns:

1.  On the proposed framework:

	a.  One of my biggest concerns about this paper is the lack of explanation of where the extended reward (and value) functions come from in Defn. 6 (and Defn. 7). If it is going to be computed by the agent, then it requires the knowledge of the reward model ($r(s,a)$ and $r_{MIN}$) and goals $g\in\mathcal{G}$ which are supposed to not be known by the agent apriori (and may never be known if the environment is large). If it is assumed to be known in advance by the agent or if the considered environments are assumed to have an extended reward function themselves (as the experiments suggest), then I believe that it is an important assumption that should perhaps be made more clear. Afterall, regular environments are not likely to have this kind of a reward function. I am curious about a discussion on this.

	b.  It would also be nice if the paper can discuss any additional limitations of the proposed framework as there seems to be no discussion on this throughout the paper.

	c.  In Sec. 3.2, it is unclear how the negation operator for a task is defined? Could the authors clarify?

	d.  Also the motivation behind redefining the negation operator for the extended value function seems to be unclear. Is this just for obtaining better bounds as stated in the paper?


2.  Regarding the experiments:

	a.  Another big concern of mine is that although the experiments support the claims of the paper, I find them to be on very limited settings of the proposed domains. For instance, in the PickOpObj domain, the agent is always in a world with only 5 objects and after it picks one, the episode immediately terminates. What about cases where, for instance, the agent is surrounded by all of the 15 objects (in a larger environment) and it has to pick up all the objects associated with a 1 in the vector $T$? In this case, for the test task $T_2$, the agent would have to sequentially pick up all the 7 objects (as there are 7 1’s in $T_2$) and then terminate the episode. The same argument can be made for the Fourrooms domain as well, where the agent would be required to pass through the locations that have 1’s in the corresponding $T$ vector. Thus, I suggest for performing the same experiments in the paper in settings where the tasks don’t terminate after just picking/reaching an object/location, but continues until all the objects/locations are picked/reached (see e.g. the tasks in [1]). I think that these additional experiments would strengthen the results by showing that the proposed framework applies to a broad range of settings.

	b.  On the same note, in my understanding, it looks as if the definition of goal-based tasks is too restrictive. For example, if there are only two yellow keys in the environment and the task is to obtain yellow keys, is the goal defined as obtaining both of the keys and then terminating or obtaining the closest one and then terminating? The experiments seem to indicate that the latter is meant by goal-based tasks. Could the authors clarify on this point and give examples.

	c.  Also, although the paper states that there have been earlier work (Saxe et al., 2017; Haarnoja et al., 2018; Van Niekerk et al., 2019; Hunt et al., 2019; Peng et al., 2019) on skill composition, there seems to be no discussion on why there is no comparison with them. If these methods are not applicable in the settings considered, or if comparison with them does not make sense, then I believe that this should explicitly be mentioned (together with the reasons) in the paper. A discussion on this would be great.


3.  Regarding related work:

	a.  Although the paper briefly talks about the GPE & GPI framework [1] in the related work section, I think there should be an extended discussion on the difference/similarities between this framework and the Boolean algebra framework developed in the paper as the two seem to be very related. For instance, by selecting the appropriate task vectors $\mathbf{w}$, one can also build a skill set that enables generalization across a distribution of tasks. In fact, this seems to have been demonstrated by a concurrent study [2].

	b.  Another closely related study on skill composition is the Option Keyboard framework [3]. It would also be nice to see a discussion on the differences/similarities of this framework and the proposed one.

	c.  I believe that providing a detailed discussion on the difference between the prior studies in 2c and the proposed method in the related work section can further clarify how the proposed method fits in the literature.


Minor comments (that have no effect on the final decision):

- Adding a newline (`\\`) to the title can make the spacing look better.

- There is a typo in the first paragraph of page 3 where “&” is used right after $\bar{r}_{MIN}$.

- In Defn. 3, I think there should be a $\min$ in the definition of $r_{M_{MIN}}$.

- In the first paragraph of Sec. 3.1, it should be “learned” instead of “said”. Also, in Defn. 5, does the agent sample a task or is a task provided by the environment? According to the experiments, the tasks seem to be coming from the environment.

- In the second paragraph of Sec. 4.1, it should be “RL method” instead of “RL learning method”.

- In Sec. 4.2, what is the value for the slip probability ($sp$)? It is not provided in any part of the paper.

- Maybe a really minor detail, but the fonts of the plots in Figure 2, 3 and 4 do not seem to match.

- In the last paragraph of Sec. 4.2, it should be $\mathcal{D}_{best}$  instead of  $\mathcal{D}_{sampled}$.

- What is EVF in Prop. 1 and Prop. 2? It should be written explicitly in its non-abbreviated form.

- In Sec. 2, is the virtual state a terminal state? If it is, I believe that naming it as so can avoid any possible confusion.

- In the second paragraph of Sec. 3.4, what does “... goal buffer according to $\mathcal{A}$.” mean? I think this should be clarified.

- Are policies assumed to be deterministic throughout the paper? If so, this should be stated explicitly.

- In Defn. 5, what does it mean to be “$\epsilon$-optimal” in task M? This should be defined explicitly.

- It seems like the $\mathcal{M}$’s in Sec. 2.1 do not correspond to the $\mathcal{M}$ in Eq. (1), as Sec. 2.1 assumes the undiscounted setting? If this is the case, I believe that the set of tasks in Sec. 2.1 should be named differently.

- In Defn. 5, does it matter whether $\mathcal{D}$ is a stationary or non-stationary distribution? (This is also assumed in Theorem 2 in Sec. 3.4) It does not seem to make any difference from the point of view of the proposed method. I believe that starting Defn. 5 as “Let D be an unknown distribution, possibly non-stationary, over a set of tasks …” would be more appropriate here.

- I believe that Eq. (1) is too general and it should be expressed more clearly. For instance, when $r_{MIN}$ and $r_{MAX}$ are both positive, all the states in $\mathcal{S}$ become desirable states and there will be no undesirable ones. When $r_{MIN} < 0$ and $r_{MAX} > 0$, some states in $\mathcal{S} \setminus \mathcal{G}$ may become desirable or undesirable states depending on $r_0(s,a)$. Having some restrictions on the possible values of $r_{MIN}$, $r_{MAX}$ and $r_0(s,a)$ can be helpful.


References:

[1] André Barreto, Shaobo Hou, Diana Borsa, David Silver, and Doina Precup. Fast reinforcement learning with generalized policy updates. Proceedings of the National Academy of Sciences, 117 (48):30079–30087, 2020.

[2] [https://openreview.net/forum?id=7IWGzQ6gZ1D](https://openreview.net/forum?id=7IWGzQ6gZ1D)

[3] Andre Barreto, Diana Borsa, Shaobo Hou, Gheorghe Comanici, Eser Aygun, Philippe Hamel, Daniel Toyama, Jonathan Hunt, Shibl Mourad, David Silver, et al. The option keyboard: combining skills in reinforcement learning. In Proceedings of the 33rd International Conference on Neural Information Processing Systems, pp. 13052–13062, 2019.


**Summary Of The Paper:**

This paper provides an interesting direction in the field of skill composition in reinforcement learning (RL). In particular, by extending Tasse et al. (2020)’s framework for Boolean task algebra, which allows for composing tasks and value functions using logical operators to yield optimal skills zero-shot, from deterministic shortest path tasks to discounted stochastic tasks, the paper proposes a new framework for lifelong RL that focuses not only on transfer between tasks for faster RL, but also gives guarantees on the performance of the agent over an unknown task distribution. Along with the framework, two main theoretical results are provided: (1) bounds on the performance of transferred policy on a new task, and (2) bounds on the necessary and sufficient number of tasks that need to be learned to guarantee performance over a distribution of tasks. Experiments are performed to verify the theoretical results in both transfer learning and lifelong RL settings in the PickUpObj and Fourrooms domains in the minigrid environment.

**Summary Of The Review:**

Overall, I think the paper can be a nice contribution to the field of lifelong RL, however, due to my concerns detailed above, I am currently voting for “marginally above the acceptance threshold”. Out of all my concerns 1a, 2a, 2b and 3a matters the most and I am willing to raise my score to accept if they are properly addressed. If, in addition, the authors can address the remaining points, this will increase the confidence in my voting to accept.

---

> ### Author Response · Authors · 2021-11-17
> **Reply to Reviewer ovqB [1/3]**
>
> Thank you for your careful review of our paper. We hope that the following points address your concerns.
>
>
> 1. On the proposed framework:
>    1. One of my biggest concerns about this paper is the lack of explanation of where the extended reward (and value) functions come from in Defn. 6 (and Defn. 7). If it is going to be computed by the agent, then it requires the knowledge of the reward model (r(s,a) and r_MIN) and goals g \in G which are supposed to not be known by the agent apriori (and may never be known if the environment is large).
>       * The extended reward and value functions are computed by the agent.  Please note that the agent receives r(s,a) for each action “a” it takes in each state “s”, so it doesn’t need a reward model. Also note that the agent stores the terminal states it has reached so far in a goal buffer, so it doesn’t need to know G in advance and instead uses the goal buffer as the estimate.
>       * Before training starts the agent only receives: the reward bounds (r_MIN, r_MAX), and the dimensionalities of the state and action space to initialise the DQN, as is standard practice.
>       * For more details, please see the pseudocode in Appendix B and the environment details in Appendix C.1.
>       * Finally, please note that just like previous works, we rely on any RL method that is able to learn goal-reaching tasks (e.g DQN) in a given environment. If the RL method is unable to reach goals in the environment of interest, then it is unable to learn the tasks.
>    2. It would also be nice if the paper can discuss any additional limitations of the proposed framework as there seems to be no discussion on this throughout the paper.
>       * One limitation of this work and previous works is the finite goal space. However, this is not a strong practical limitation since you can think of G as the set of latent goal states in a partially observable setting. For example, when the goal states are defined as continuous regions of the state space, the goal observations may not be goal states but the latent goal states are usually still finite. In this case one need only learn a map from the goal observations to the latent goal states (e.g with a supervised or clustering approach). This is a promising experimental extension for future work.
>       * Just like previous works, we inherit some of the problems in regular RL. SOPGOL relies on an RL method that is able to learn goal-reaching tasks (e.g DQN). If the RL method is unable to reach goals in the environment of interest, then it is unable to learn the tasks and SOPGOL will be unable to learn the task vectors. An interesting direction for future work will be to combine SOPGOL with methods that attempt to address the delayed reward problem (e.g with reward shaping or subgoal temporal compositions depending on the setting). We will include a discussion of both of these points in an updated version of the paper
>    3. In Sec. 3.2, it is unclear how the negation operator for a task is defined? Could the authors clarify?
>       * The intuition behind the re-defined negation operator in Sec. 3.2 is as follows: since each goal is either desirable or not, the optimal Q(s,g,a) is either Q_MAX(s,g,a) or Q_MIN(s,g,a). Hence, if Q(s,g,a) is closer to Q_MIN(s,g,a), then its negation should be Q_MAX(s,g,a), and vice-versa.
>    4. Also the motivation behind redefining the negation operator for the extended value function seems to be unclear. Is this just for obtaining better bounds as stated in the paper?
>       * Yes, which is really important because we want to lose as little optimality as possible. Tasse et al. 2020 only considered optimal value functions in their theoretical results.

---

> > ### Author Response · Authors · 2021-11-17
> > **Reply to Reviewer ovqB [2/3]**
> >
> >
> > 2. Regarding the experiments:
> >    1. The experiments are on very limited settings of the proposed domains. For example, what about the case where the agent needs to sequentially pickup desirable objects like in [1]. I suggest for performing the same experiments in the paper in settings where the tasks don’t terminate after just picking/reaching an object/location, but continues until all the objects/locations are picked/reached (see e.g. the tasks in [1]).
> >       * We used these tasks because they illustrate what we care about in this work. That is many-goal tasks in an environment with a large goal space, where the agent needs to optimally reach a desirable goal.
> >       * The example you gave can be rephrased as “repeatedly pick up desirable objects”. Notice that this is a temporal task. In this case, the trained agent can simply be allowed to continue picking up objects by not transitioning to the terminal state upon picking up an object. See [4] (Sec 5.4, Fig 5) for an example of this, since we are using the exact same definition of goals as them.
> >       * While we do not consider temporal compositions in this work, it is definitely an interesting avenue for future work.
> >    2. On the same note, in my understanding, it looks as if the definition of goal-based tasks is too restrictive. For example, if there are only two yellow keys in the environment and the task is to obtain yellow keys, is the goal defined as obtaining both of the keys and then terminating or obtaining the closest one and then terminating? The experiments seem to indicate that the latter is meant by goal-based tasks.
> >       * Indeed, goal-based tasks here mean reaching the closest goal and then terminating. That is the same as most previous works (Abel et al. 2018; Tasse et al. 2020; Van Niekerk et al., 2019; Saxe et al., 2017; Haarnoja et al., 2018; Hunt et al., 2019; Peng et al., 2019). The idea here is that in goal-reaching tasks, we want the agent to optimally reach a desirable goal. Relating to the previous comment, temporal tasks would involve reaching sequences of desirable goals.
> >    3. Although the paper states that there have been earlier work (Saxe et al., 2017; Haarnoja et al., 2018; Van Niekerk et al., 2019; Hunt et al., 2019; Peng et al., 2019) on skill composition, there seems to be no discussion on why there is no comparison with them
> >       * These works, including Tasse et al. 2020, address the question of how to compose learned skills to produce new ones. They do not address the question of how to leverage such compositions to tackle the lifelong RL problem. For example, Tasse et al. 2020 assume that the expression for the task compositions is given and also only considers zero-shot composition of optimal value functions. We build on top of Tasse et al. 2020 (instead of the other works) because they provide a formal framework for doing logical compositions of skills, while the other works only consider unions and intersections. Note that we also compare our approach with Abel et al. 2018 in Sec. 4.2. It is an efficient LRL approach and illustrates the problem when there is no compositional generalisation (Fig. 4).

---

> > > ### Author Response · Authors · 2021-11-17
> > > **Reply to Reviewer ovqB [3/3]**
> > >
> > >
> > > 3. Regarding related work
> > >    1. Although the paper briefly talks about the GPE & GPI framework [1] in the related work section, I think there should be an extended discussion on the difference/similarities between this framework and the Boolean algebra framework developed in the paper as the two seem to be very related. For instance, by selecting the appropriate task vectors w, one can also build a skill set that enables generalization across a distribution of tasks. In fact, this seems to have been demonstrated by a concurrent study [2].
> > >       * Note that these works [1,2] are all based on the successor features with GPI (SF+GPI) results of [5]. As mentioned in Section 3.3, the way we extract the policies from the EVFs is similar to GPI with stronger bounds. See the comparison in Appendix A.4. We also use the disjunction of the new and composed skill to speed up learning when the new task is not fully expressible in terms of past tasks. This is also similar to GPI (see Section 3.4 with footnote 4).
> > >       * As for works based on SFs, note that SFs make the very important assumption that the task rewards are well approximated linearly by a set of features. Unfortunately, that is not true for many-goal tasks where goals are either desirable or not. For example, consider the simple case where the environment has only 4 goals (like in [5]). Since [5] shows that features are equivalent to rewards, let us say [0,0,1,1] and [0,1,0,1] are the features. One can see that there is no vector w = [w1, w2] that gives a good approximation for the new task [0,1,1,0] (i.e their exclusive OR).
> > >       * Additionally, even when the features are handpicked to span the task space, one still needs logical composition to get the skill for [0,1,1,0] from those of [0,0,1,1] and [0,1,0,1] (since GPI is similar to just a disjunction)
> > >       * SFs are suitable for settings where the current task is defined by preferences over a set tasks (the vector w is the preferences). This makes it suitable for temporal compositions like in [3] where the preferences change over time. An interesting avenue for future work would be to combine SFs with this work to solve new tasks that are preferences over the logical combinations of past tasks. We can add a discussion of this to the paper.
> > >    2. Another closely related study on skill composition is the Option Keyboard framework [3]. It would also be nice to see a discussion on the differences/similarities of this framework and the proposed one
> > >       * This work is also based on [5]. Additionally, it mainly focuses on temporal compositions. An interesting avenue for future work would be to combine our work with temporal compositions like this one or other options and HRL approaches. Thanks for the suggestions regarding the related works, we will incorporate them into the updated paper.
> > >
> > >
> > > [4] B. Van Niekerk et al. 2019; Composing value functions in reinforcement learning.
> > >
> > > [5] Andre Barreto et al. 2018; Transfer in deep reinforcement learning using successor features and generalised policy improvement

---

> > > > ### Comment · Reviewer_ovqB · 2021-11-22
> > > > **Response to the Authors**
> > > >
> > > > I would like to thank the authors for their response. However, I still have some concerns that I think were not successfully addressed:
> > > >
> > > >   1. Thank you for successfully addressing the concerns on the proposed framework. However, I believe that adding a discussion on 1b on the revised version of the paper would definitely be helpful for the reader in understanding the limitations of the proposed framework. The conclusion section is probably the best place to include it.
> > > >
> > > >   2. Thank you for successfully addressing the concerns on the experiments. However, I still suggest for a discussion about how the proposed framework would be able to handle “temporal tasks” on the revised version of the paper.
> > > >
> > > >   3. I am not sure about how the linearity assumption of successor features prevents them from handling many-goal tasks. Considering an environment with 4 goals one can select the task vectors as $w_1= [1, -1, -1, -1]$, $w_2= [-1, 1, -1, -1]$, $w_3= [-1,-1, 1, -1]$ and $w_4= [-1, -1, -1, 1]$, and after learning the policies associated with these tasks, the resulting GPI policy would be able to solve all possible upcoming tasks. This is of course when the features are in a one-hot form. The concurrent study [2] that I provided above does this exactly. Thus, I still believe that the related work section should be extended to include a discussion about the current study and how [1] would be able to tackle the same problem. I also expect [3] to be discussed.

---

> > > > > ### Comment · Reviewer_ovqB · 2021-11-26
> > > > > **Final Review**
> > > > >
> > > > > Dear authors, thanks for all the explanations and clarifications. Most of my concerns have been resolved. I have increased my score to 8. However, I would like to note that I still have some concerns on how the proposed framework would be able to handle “temporal” tasks. There seems to be no discussion on this in the revised version of the paper. If accepted, I would still encourage the authors to add a discussion. Of course, performing experiments with these tasks would even be greater.

---

> > > > > > ### Author Response · Authors · 2021-11-26
> > > > > > **Thank you for the score increase**
> > > > > >
> > > > > > We will definitely add the discussion on temporal tasks in the updated paper.

---

### Official Review · Reviewer_BUDa · 2021-11-02

**Correctness:** 4
**Technical Novelty And Significance:** 2
**Empirical Novelty And Significance:** 3
**Recommendation:** 6
**Confidence:** 3

**Main Review:**

Good work on this paper. I appreciate the background session as I had forgotten lots of details from the boolean task algebra, since I read it. The papar is very sound and the concern about the number of skills is reassuring. Using GPI on the extended Q_SOP and extended Q~ is a useful trick and yields good results. Within the problem formulation, the empirical method works well enough.

Some improvements can be made on the paper though:

1. The following claim right before theorem 2 seems incorrect: ".We now show ... generalises over any unknown non-stationary task distribution after learning only a number of tasks logarithmic in the size of the task space". My understanding is that theorem 2 is actually a lower bound on the number of goals needed to learn, and the upper bound is |G|, unless we make further assumptions, as you have shown already with D_worst in the experiments. The same applies to similar claims around the paper, for example, "which are
sub-logarithmic in the size of the task space" (abstract),

2. The background session 2.1. is missing the Lemma 1 from previous work which is crucial to building its intuition and recover the non-extended reward and value (i.e. just maximize over G to recover the non-extended quantities). Without it, readers unfamiliar with previous work may struggle to understand definition 3 onwards.

3. I miss some discussion on the limitations of the method, specially in terms of using finite set of goals, which doesn't seem to be a strong requirement of the related work discussed. It's often not feasible to do that in some scenarios, so it can hinder the further applicability of this method, for example, some goals are defined as continuous regions of the state space.


**Summary Of The Paper:**

This paper builds upon previous work on the Boolean Task Algebra For Reinforcement Learning extending it to the discounted and stochastic tasks and to the lifelong RL setup. The authors show that techniques introduce in the previous work can perform close to optimal in a zero-shot transfer scenario. Next, they look into the number of skills needed in order to solve all the tasks and arrive at some lower bounds which are encouraging since they are close to log|G|, this is somewhat supported by empirical analysis later on.
The authors also introduce a method for using the known skills with SOP to solve new tasks and to bootstrapping the learning of new skills if needed. Overall, the method is very quick to learn when compared to a simple Q-learning baseline. This work is orthogonal to lots of related ones in the skill learning literature and would combine well with many of them.

**Summary Of The Review:**

This is a good paper with a sound method and good results. The author just need to work on a few points to award a clear acceptance.
Some claims about the efficiency of the algorithm with the number of goals need to be removed. A important step of the background is missing. Finally, some short discussion about limitations would be welcome.

---

> ### Author Response · Authors · 2021-11-14
> **Reply to Reviewer BUDa**
>
> Thank you for your careful review of our paper and are glad you liked it. We hope that the following points address your concerns. Please let us know if there is anything else that we can clarify or improve.
>
>
> 1. The following claim right before theorem 2 seems incorrect: ".We now show ... generalises over any unknown non-stationary task distribution after learning only a number of tasks logarithmic in the size of the task space". My understanding is that theorem 2 is actually a lower bound on the number of goals needed to learn, and the upper bound is |G|
>    * Theorem 2 bounds the number of tasks that need to be learned. The lower bound is \ceil{log{|G|}} because that is the minimum number of tasks that span the task space, as per Tasse et al 2020 and as can be seen in Table 1 for example. The upper bound is |G| because that is the dimensionality of the task space. This is logarithmic in the size of the task space since the number of tasks is 2^|G|. We will make that clearer in the paper.
> 2. The background session 2.1. is missing the Lemma 1 from previous work which is crucial to building its intuition and recover the non-extended reward and value (i.e. just maximize over G to recover the non-extended quantities).
>    * Please see the last sentence in the paragraph following Def 2. We are happy to add Lemma 1 from Tasse et al. to improve clarity.
> 3. I miss some discussion on the limitations of the method, specially in terms of using finite set of goals, which doesn't seem to be a strong requirement of the related work discussed. It's often not feasible to do that in some scenarios, so it can hinder the further applicability of this method, for example, some goals are defined as continuous regions of the state space.
>    * The finite goal space is indeed a limitation of this work. However, to the best of our knowledge all previous works (e.g [1,2]) only consider finite (latent) goals (see their experiments). You can think of G as the set of latent goal states in a partially observable setting. For example, when the goal states are defined as continuous regions of the state space, the goal observations may not be goal states but the latent goal states are usually still finite. In this case one need only learn a map from the goal observations to the latent goal states (e.g with a supervised or clustering approach). This is definitely a promising experimental extension for future work.
>    * Just like previous works, we inherit some of the problems in regular RL. SOPGOL relies on an RL method that is able to learn goal-reaching tasks (e.g DQN). If the RL method is unable to reach goals in the environment of interest, then it is unable to learn the tasks and SOPGOL will be unable to learn the task vectors. An interesting direction for future work will be to combine SOPGOL with methods that attempt to address the delayed reward problem (e.g with reward shaping or subgoal temporal compositions depending on the setting).
>
>
> [1] D. Abel et al., Policy and value transfer in lifelong reinforcement learning.
> [2] Andre Barreto et al., Transfer in deep reinforcement learning using successor features and generalised policy improvement.

---

> > ### Author Response · Authors · 2021-11-26
> > **Thanks for your suggestions**
> >
> > We have updated the paper to reflect all of your suggestions. As we are nearing the end of the discussion period, do you have any outstanding concerns? We are happy to address them.

---

### Official Review · Reviewer_td5N · 2021-11-03

**Correctness:** 4
**Technical Novelty And Significance:** 3
**Empirical Novelty And Significance:** 2
**Recommendation:** 6
**Confidence:** 3

**Main Review:**

This work is sound and can potentially shape future lifelong reinforcement learning (LRL) research.
I'm a bit skeptical however about its applicability in any use cases.

**Strengths:**
- The introduced problem formulation is a good first step towards lifelong compositional RL agents.
- the theoretical guarantees for SOGPOL are satisfying
- the empirical study is well-aligned w/ the problem formulation and highlights the relevant regimes , i.e., 4.1 a)  vs 4.1 b) vs 4.2

**Weaknesses:**
- The boolean task algebra is quite restrictive. I don't see many use cases for it. If the authors introduce a new problem setting, they should motivate it.
- assuming the agent can interact with the task as long as it needs is quite an unrealistic assumption in LRL.
- all experiments on toy tasks. I think a more realistic benchmark should be introduced.
- there's no LRL baselines.
- in SOGPOL,
    - the new skill is initialize randomly. This will be pretty inefficient is more realistic settings.
    - IIUC, when learning, to enable transfer from previous tasks, goals need to be reached. But in realistic settings, goals will be hard to reach. So essentially lots time and compute will be wasted relearning over and over same things.

**Minor details:**
- def 3: $r_{MAX}$ is the same as $r_{MIN}$
- theorem 1: what is $\gamma$ ?
- what's the difference between prop. 1 and prop. 2 ?
- theorem 2: can you provide some intuition/explanation on those bounds?

**Summary Of The Paper:**

This works introduces a lifelong RL problem in which new tasks can be possibly be expressed as a logical composition of previous ones.

Accordingly, they introduce algorithm (SOPGOL) that can autonomously determine whether a new task can be immediately solved using its existing abilities, or whether a task-specific skill should be learned.

The authors provide some theoretical guarantees on the performance of the algorithm, as well as empirical evidence that it can work in a toy setting.

**Summary Of The Review:**

I think this work as merits.
I'm giving a weak reject because I would like to see more motivation for this work and how it could be adapted to use cases.
I'm currently keeping my confidence score somewhat low because the theoretical contribution could be significant enough to alleviate the need for thorough motivations/experiments.

======= POST REBUTTAL ==============

Thanks for all the explanations and clarification.
Some of my concerns have been resolved.
I'll increase my score to 6.
I will not go higher because I still think some experiments should be added in a realistic lifelong RL benchmark, i.e., where the tasks won't actually obey the theoretical task algebra. Upon acceptance, I still encourage the authors to add such an experiment.

Minor detail: please add the provided intuitions for Theorem 2 in the manuscript

---

> ### Author Response · Authors · 2021-11-14
> **Reply to Reviewer td5N [1/2]**
>
> Thank you for your careful review of our paper. We hope that the following points address your concerns.
>
>
> 1. The boolean task algebra is quite restrictive. I don't see many use cases for it. If the authors introduce a new problem setting, they should motivate it.
>    * We consider the same problem setting as previous work [1], slightly extended to highlight the generalisation issue. Please see Section 3.1 for the motivation.
>    * At a high level, the motivation for this is if we are ever going to have lifelong robots acting in the real world to solve tasks we care about, then they need at the very least to be able to solve many goal tasks (since real-world tasks are seldom single-goal as has been historically considered in most RL works). Consider for example a domestic robot that needs to cook various types of food, buy groceries, deliver items, clean parts of the house, etc. Hence we care about agents in an environment that receive various many-goal tasks through-out their lifetime.
>    * Since we care about sample efficiency and generalisation ability, we leverage the boolean task algebra framework since it gives agents a combinatorial number of skills. Agents can then leverage past skills to solve combinatorial tasks like “buy groceries or deliver water and food to the construction site”.
> 2. Assuming the agent can interact with the task as long as it needs is quite an unrealistic assumption in LRL.
>    * Please note that we make no such assumption. As is common in RL, we use the epsilon-optimal formulation just like previous works [1,2]. Hence we can stop training the agent after n episodes (as we do in our experiments), and the resulting Q-values will be epsilon-optimal for some epsilon (which is the same for any other stopping condition).
> 3. All experiments on toy tasks. I think a more realistic benchmark should be introduced.
>    * We used the PickUpObj domain because it illustrates what we care about in this work. It is an environment with many-goal tasks which can be learned with an off-the-shelve RL algorithm like deep-Q learning (but in a very sample inefficient way and with poor generalisation over the task distribution).
>    * Notice that the complexity of the environment is not relevant for this work, since that only affects how long the regular RL algorithm takes to learn it. In all our experiments we used the same RL algorithm in SOPGOL and the baseline.
>    * We are happy to redo the experiments with any other environment you think adds more to the story while having many-goal tasks which are solvable with an off-the-shelve RL algorithm.
> 4. There's no LRL baselines.
>    *  Please see section 4.2, where we use the maxQ baseline from [1]. This has been shown by previous works [1,2] to be an effective approach.
>    * We are happy to include or discuss any other baseline you feel is missing.
> 5. In SOPGOL, the new skill is initialize randomly. This will be pretty inefficient is more realistic settings.
>    * Our aim here is to address transfer and generalisation via composition. As discussed in the paper, this can be combined with other transfer approaches for faster learning. E.g One can initialize the Q-values with maxQ.
> 6. In SOPGOL, IIUC, when learning, to enable transfer from previous tasks, goals need to be reached. But in realistic settings, goals will be hard to reach. So essentially lots time and compute will be wasted relearning over and over same things.
>    * This is a problem with regular RL. SOPGOL relies on an RL method that is able to learn tasks (e.g DQN). If the RL method is unable to reach goals, then it is unable to learn the tasks irrespective of SOPGOL. If the RL method is able to reach goal states, then the task vector can be learned much faster than learning to solve the task. SOPGOL leverages this fact together with composition to enable faster policy learning of new tasks that are similar to past tasks, and no policy learning is needed when the task is in fact fully expressible in terms of past tasks. In this way, we actually require *less* time and compute in solving new tasks.
>
>
> [1] D. Abel et al., Policy and value transfer in lifelong reinforcement learning.
>
> [2] Andre Barreto et al., Transfer in deep reinforcement learning using successor features and generalised policy improvement.

---

> > ### Author Response · Authors · 2021-11-14
> > **Reply to Reviewer td5N [2/2]**
> >
> > 7. def 3: $r_{M_{MIN}}$  is the same as $ r_{M_{MAX}}$
> >    * Thanks for noticing the typo. $r_{M_{MIN}}$ should be using $\min_{M \in \mathcal{M}}$.
> > 8. theorem 1: what is $\gamma$?
> >    * The standard discount factor. Please see the definition of an MDP in the background section (Sec 2).
> > 9. what's the difference between prop. 1 and prop. 2 ?
> >    * Prop 2 is for discounted goal-reaching tasks (Equation 1), and uses our new definition for the negation operator. For more clarity, we are happy to use a different symbol for the set of tasks in prop 1 and Sec 2.1 in general.
> > 10. theorem 2: can you provide some intuition/explanation on those bounds?
> >    * Theorem 2 bounds the number of tasks that need to be learned. The lower bound is $\lceil log{|\mathcal{G}| \rceil }$ because that is the minimum number of tasks that span the task space, as per Tasse et al 2020 and as can be seen in Table 1 for example. The upper bound is $|\mathcal{G}|$ because that is the dimensionality of the goal space. Since the number of tasks is $|\mathcal{M}| = 2^{|\mathcal{G}|}$, we have that the upper bound $|\mathcal{G}| = log|\mathcal{M}|$ is logarithmic in the size of the task space.

---

> > > ### Author Response · Authors · 2021-11-25
> > > **Thanks for the score update**
> > >
> > > We have included the explanation for Theorem 2's bound in the revised paper.
> > >
> > > > I still think some experiments should be added in a realistic lifelong RL benchmark, i.e., where the tasks won't actually obey the theoretical task algebra.
> > >
> > > We are curious as to the kinds of tasks that you believe we should cover (a specific example would be great).

---

### Author Response · Authors · 2021-11-23
**Paper Revision**

We thank all the reviewers for their valuable comments and suggestions. We have updated the paper with those suggestions and have included our clarifications for the various concerns. We have also highlighted all the main changes in blue for your convenience.

---

### Decision · Program_Chairs · 2022-01-20

**Decision:**

Accept (Poster)

**Comment:**

I thank the authors for their submission and active participation in the discussions. This papers is borderline. On the positive side, reviewers emphasized this is a well written [ovqB,1zPe] and sound paper [BUDa] with good theoretical [td5N,ovqB,1zPe] and empirical [BUDa,td5N,ovqB] results. On the negative side, reviewers remarked clarity [KyZj,AVki], incremental with respect to Tasse et al (2020) [KyZj], relatively restricted Boolean task algebra [td5N], toyish nature of the environments considered [ovqB], and some missing details [1zPe]. During discussion, the sentiment seems to be somewhat lukewarm with none of the reviewers strongly favoring acceptance or rejection. It seems the main remaining concern is around the toyish nature of the environments used in this paper. I acknowledge that and I believe the authors could include experiments on more complex environments. However, I also give the authors credit for addressing most of the reviewer's concerns during rebuttal and for presenting a solid empirical and theoretical result that the research community can build upon in the future. I am therefore recommending acceptance of this paper and highly encourage the authors to further improve their paper based on the reviewer feedback.